# An Efficient Tester-Learner for Halfspaces

## Abstract

We give the first efficient algorithm for learning halfspaces in the testable learning model recently defined by Rubinfeld and Vasilyan [RV23]. In this model, a learner certifies that the accuracy of its output hypothesis is near optimal whenever the training set passes an associated test, and training sets drawn from some target distribution must pass the test. This model is more challenging than distribution-specific agnostic or Massart noise models where the learner is allowed to fail arbitrarily if the distributional assumption does not hold. We consider the setting where the target distribution is the standard Gaussian in $d$ dimensions and the label noise is either Massart or adversarial (agnostic). For Massart noise, our tester-learner runs in polynomial time and outputs a hypothesis with (information-theoretically optimal) error $\mathsf{opt} + \epsilon$ (and extends to any fixed strongly log-concave target distribution). For adversarial noise, our tester-learner obtains error $O(\mathsf{opt}) + \epsilon$ in polynomial time. Prior work on testable learning ignores the labels in the training set and checks that the empirical moments of the covariates are close to the moments of the base distribution. Here we develop new tests of independent interest that make critical use of the labels and combine them with the moment-matching approach of [GKK23]. This enables us to implement a testable variant of the algorithm of [DKTZ20a, DKTZ20b] for learning noisy halfspaces using nonconvex SGD.

## 1  Introduction

Learning halfspaces in the presence of noise is one of the most basic and well-studied problems in computational learning theory. A large body of work has obtained results for this problem under a variety of different noise models and distributional assumptions (see e.g. [BH21] for a survey). A major issue with common distributional assumptions such as Gaussianity, however, is that they can be hard or impossible to verify in the absence of any prior information.

The recently defined model of testable learning [RV23] addresses this issue by replacing such assumptions with efficiently testable ones. In this model, the learner is required to work with an arbitrary input distribution $D_{\mathcal{X}\mathcal{Y}}$ and verify any assumptions it needs to succeed. It may choose to reject a given training set, but if it accepts, it is required to output a hypothesis with error close to $\mathsf{opt}(\mathcal{C}, D_{\mathcal{X}\mathcal{Y}})$, the optimal error achievable over $D_{\mathcal{X}\mathcal{Y}}$ by any function in a concept class $\mathcal{C}$. Further, whenever the training set is drawn from a distribution $D_{\mathcal{X}\mathcal{Y}}$ whose marginal is truly a well-behaved target distribution $D^*$ (such as the standard Gaussian), the algorithm is required to accept with high probability. Such an algorithm, or tester-learner, is then said to testably learn $\mathcal{C}$ with respect to target marginal $D^*$. (See Definition 2.1.) Note that unlike ordinary distribution-specific agnostic learners, a tester-learner must take some nontrivial action *regardless* of the input distribution.

The work of [RV23, GKK23] established foundational algorithmic and statistical results for this model and showed that testable learning is in general provably harder than ordinary distribution-specific agnostic learning. As one of their main algorithmic results, they showed tester-learners for

the class of halfspaces over $\mathbb{R}^d$ that succeed whenever the target marginal is Gaussian (or one of a more general class of distributions), achieving error $\mathsf{opt} + \epsilon$ in time and sample complexity $d^{\widetilde{O}(1/\epsilon^2)}$. This matches the running time of ordinary distribution-specific agnostic learning of halfspaces over the Gaussian using the standard approach of [KKMS08]. Their testers are simple and label-oblivious, and are based on checking whether the low-degree empirical moments of the unknown marginal match those of the target $D^*$.

These works essentially resolve the question of designing tester-learners achieving error $\mathsf{opt} + \epsilon$ for halfspaces, matching known hardness results for (ordinary) agnostic learning [GGK20, DKZ20, DKPZ21]. Their running time, however, necessarily scales exponentially in $1/\epsilon$.

A long line of research has sought to obtain more efficient algorithms at the cost of relaxing the optimality guarantee [ABL17, DKS18, DKTZ20a, DKTZ20b]. These works give polynomial-time algorithms achieving bounds of the form $\mathsf{opt} + \epsilon$ and $O(\mathsf{opt}) + \epsilon$ for the Massart and agnostic setting respectively under structured distributions (see Section 1.1 for more discussion). The main question we consider here is whether such guarantees can be obtained in the testable learning framework.

**Our contributions.** In this work we design the first tester-learners for halfspaces that run in fully polynomial time in all parameters. We match the optimality guarantees of fully polynomial-time learning algorithms under Gaussian marginals for the Massart noise model (where the labels arise from a halfspace but are flipped by an adversary with probability at most $\eta$) as well as for the agnostic model (where the labels can be completely arbitrary). In fact, for the Massart setting our guarantee holds with respect to any chosen target marginal $D^*$ that is isotropic and strongly log-concave, and the same is true of the agnostic setting albeit with a slightly weaker guarantee.

**Theorem 1.1** (Formally stated as Theorem 4.1). *Let $\mathcal{C}$ be the class of origin-centered halfspaces over $\mathbb{R}^d$, and let $D^*$ be any isotropic strongly log-concave distribution. In the setting where the labels are corrupted with Massart noise at rate at most $\eta < \frac{1}{2}$, $\mathcal{C}$ can be testably learned w.r.t. $D^*$ up to error $\mathsf{opt} + \epsilon$ using $\mathrm{poly}(d, \frac{1}{\epsilon}, \frac{1}{1-2\eta})$ time and sample complexity.*

**Theorem 1.2** (Formally stated as Theorem 5.1). *Let $\mathcal{C}$ be as above. In the adversarial noise or agnostic setting where the labels are completely arbitrary, $\mathcal{C}$ can be testably learned w.r.t. $\mathcal{N}(0, I_d)$ up to error $O(\mathsf{opt}) + \epsilon$ using $\mathrm{poly}(d, \frac{1}{\epsilon})$ time and sample complexity.*

**Our techniques.** The tester-learners we develop are significantly more involved than prior work on testable learning. We build on the nonconvex optimization approach to learning noisy halfspaces due to [DKTZ20a, DKTZ20b] as well as the structural results on fooling functions of halfspaces using moment matching due to [GKK23]. Unlike the label-oblivious, global moment tests of [RV23, GKK23], our tests make crucial use of the labels and check *local* properties of the distribution in regions described by certain candidate vectors. These candidates are approximate stationary points of a natural nonconvex surrogate of the 0-1 loss, obtained by running gradient descent. When the distribution is known to be well-behaved, [DKTZ20a, DKTZ20b] showed that any such stationary point is in fact a good solution (for technical reasons we must use a slightly different surrogate loss). Their proof relies crucially on structural geometric properties that hold for these well-behaved distributions, an important one being that the probability mass of any region close to the origin is proportional to its geometric measure.

In the testable learning setting, we must efficiently check this property for candidate solutions. Since these regions may be described as intersections of halfspaces, we may hope to apply the moment-matching framework of [GKK23]. Naïvely, however, they only allow us to check in polynomial time that the probability masses of such regions are within an additive constant of what they should be under the target marginal. But we can view these regions as sub-regions of a known band described by our candidate vector. By running moment tests on the distribution *conditioned* on this band and exploiting the full strength of the moment-matching framework, we are able to effectively convert our weak additive approximations to good multiplicative ones. This allows us to argue that our stationary points are indeed good solutions.

**Limitations and Future Work.** In this paper we provide the first efficient tester-learners for halfspaces when the noise is either adversarial or Massart. An interesting direction for future work would be to design tester-learners for the agnostic setting whose target marginal distributions may lie within a large family (e.g., strongly log-concave distributions) but still achieve error of $O(\mathsf{opt})$. Another interesting direction is providing tester-learners that are not tailored to a single target distribution, but are guaranteed to accept any member of a large family of distributions.

## 1.1 Related work

We provide a partial summary of some of the most relevant prior and related work on efficient algorithms for learning halfspaces in the presence of adversarial label or Massart noise, and refer the reader to [BH21] for a survey.

In the distribution-specific agnostic setting where the marginal is assumed to be isotropic and log-concave, [KLS09] showed an algorithm achieving error $O(\mathsf{opt}^{1/3}) + \epsilon$ for the class of origin-centered halfspaces. [ABL17] later obtained $O(\mathsf{opt}) + \epsilon$ using an approach that introduced the principle of iterative *localization*, where the learner focuses attention on a band around a candidate halfspace in order to produce an improved candidate. [Dan15] used this principle to obtain a PTAS for agnostically learning halfspaces under the uniform distribution on the sphere, and [BZ17] extended it to more general $s$-concave distributions. Further works in this line include [YZ17, Zha18, ZSA20, ZL21]. [DKTZ20b] introduced the simplest approach yet, based entirely on nonconvex SGD, and showed that it achieves $O(\mathsf{opt}) + \epsilon$ for origin-centered halfspaces over a wide class of structured distributions. Other related works include [DKS18, DKTZ22].

In the Massart noise setting with noise rate bounded by $\eta$, work of [DGT19] gave the first efficient distribution-free algorithm achieving error $\eta + \epsilon$; further improvements and followups include [DKT21, DTK22]. However, the optimal error opt achievable by a halfspace may be much smaller than $\eta$, and it has been shown that there are distributions where achieving error competitive with opt as opposed to $\eta$ is computationally hard [DK22, DKMR22]. As a result, the distribution-specific setting remains well-motivated for Massart noise. Early distribution-specific algorithms were given by [ABHU15, ABHZ16], but a key breakthrough was the nonconvex SGD approach introduced by [DKTZ20a], which achieved error $\mathsf{opt} + \epsilon$ for origin-centered halfspaces efficiently over a wide range of distributions. This was later generalized by [DKK$^+$22].

## 1.2 Technical overview

Our starting point is the nonconvex optimization approach to learning noisy halfspaces due to [DKTZ20a, DKTZ20b]. The algorithms in these works consist of running SGD on a natural non-convex surrogate $\mathcal{L}_\sigma$ for the 0-1 loss, namely a smooth version of the ramp loss. The key structural property shown is that if the marginal distribution is structured (e.g. log-concave) and the slope of the ramp is picked appropriately, then any $\mathbf{w}$ that has large angle with an optimal $\mathbf{w}^*$ cannot be an approximate stationary point of the surrogate loss $\mathcal{L}_\sigma$, i.e. that $\|\nabla\mathcal{L}_\sigma(\mathbf{w})\|$ must be large. This is proven by carefully analyzing the contributions to the gradient norm from certain critical regions of $\mathrm{span}(\mathbf{w}, \mathbf{w}^*)$, and crucially using the distributional assumption that the probability masses of these regions are proportional to their geometric measures. (See Fig. 3.) In the testable learning setting, the main challenge we face in adapting this approach is checking such a property for the unknown distribution we have access to.

A preliminary observation is that the critical regions of $\mathrm{span}(\mathbf{w}, \mathbf{w}^*)$ that we need to analyze are rectangles, and are hence functions of a small number of halfspaces. Encouragingly, one of the key structural results of the prior work of [GKK23] pertains to "fooling" such functions. Concretely, they show that whenever the true marginal $D_\mathcal{X}$ matches moments of degree at most $\widetilde{O}(1/\tau^2)$ with a target $D^*$ that satisfies suitable concentration and anticoncentration properties, then $|\mathbb{E}_{D_\mathcal{X}}[f] - \mathbb{E}_{D^*}[f]| \leq \tau$ for any $f$ that is a function of a small number of halfspaces. If we could run such a test and ensure that the probabilities of the critical regions over our empirical marginal are also related to their areas, then we would have a similar stationary point property.

However, the difficulty is that since we wish to run in fully polynomial time, we can only hope to fool such functions up to $\tau$ that is a constant. Unfortunately, this is not sufficient to analyze the probability masses of the critical regions we care about as they may be very small.

The chief insight that lets us get around this issue is that each critical region $R$ is in fact of a very specific form, namely a rectangle that is axis-aligned with $\mathbf{w}$: $R = \{\mathbf{x} : \langle\mathbf{w}, \mathbf{x}\rangle \in [-\sigma, \sigma]$ and $\langle\mathbf{v}, \mathbf{x}\rangle \in [\alpha, \beta]\}$ for some values $\alpha, \beta, \sigma$ and some $\mathbf{v}$ orthogonal to $\mathbf{w}$. Moreover, we *know* $\mathbf{w}$, meaning we can efficiently estimate the probability $\mathbb{P}_{D_\mathcal{X}}[\langle\mathbf{w}, \mathbf{x}\rangle \in [-\sigma, \sigma]]$ up to constant multiplicative factors without needing moment tests. Denoting the band $\{\mathbf{x} : \langle\mathbf{w}, \mathbf{x}\rangle \in [-\sigma, \sigma]\}$ by $T$ and writing $\mathbb{P}_{D_\mathcal{X}}[R] = \mathbb{P}_{D_\mathcal{X}}[\langle\mathbf{v}, \mathbf{x}\rangle \in [\alpha, \beta] \mid \mathbf{x} \in T]\mathbb{P}_{D_\mathcal{X}}[T]$, it turns out that we should expect $\mathbb{P}_{D_\mathcal{X}}[\langle\mathbf{v}, \mathbf{x}\rangle \in [\alpha, \beta] \mid \mathbf{x} \in T] = \Theta(1)$, as this is what would occur under the structured target distri-

bution $D^*$. (Such a "localization" property is also at the heart of the algorithms for approximately learning halfspaces of, e.g., [ABL17, Dan15].) To check this, it suffices to run tests that ensure that $\mathbb{P}_{D_\mathcal{X}}[\langle \mathbf{v}, \mathbf{x} \rangle \in [\alpha, \beta] \mid \mathbf{x} \in T]$ is within an additive constant of this probability under $D^*$.

We can now describe the core of our algorithm (omitting some details such as the selection of the slope of the ramp). First, we run SGD on the surrogate loss $\mathcal{L}$ to arrive at an approximate stationary point and candidate vector $\mathbf{w}$ (technically a list of such candidates). Then, we define the band $T$ based on $\mathbf{w}$, and run tests on the empirical distribution conditioned on $T$. Specifically, we check that the low-degree empirical moments conditioned on $T$ match those of $D^*$ conditioned on $T$, and then apply the structural result of [GKK23] to ensure conditional probabilities of the form $\mathbb{P}_{D_\mathcal{X}}[\langle \mathbf{v}, \mathbf{x} \rangle \in [\alpha, \beta] \mid \mathbf{x} \in T]$ match $\mathbb{P}_{D^*}[\langle \mathbf{v}, \mathbf{x} \rangle \in [\alpha, \beta] \mid \mathbf{x} \in T]$ up to a suitable additive constant. This suffices to ensure that even over our empirical marginal, the particular stationary point $\mathbf{w}$ we have is indeed close in angular distance to an optimal $\mathbf{w}^*$.

A final hurdle that remains, often taken for granted under structured distributions, is that closeness in angular distance $\measuredangle(\mathbf{w}, \mathbf{w}^*)$ does not immediately translate to closeness in terms of agreement, $\mathbb{P}[\text{sign}(\langle \mathbf{w}, \mathbf{x} \rangle) \neq \text{sign}(\langle \mathbf{w}^*, \mathbf{x} \rangle)]$, over our unknown marginal. Nevertheless, we show that when the target distribution is Gaussian, we can run polynomial-time tests that ensure that an angle of $\theta = \measuredangle(\mathbf{w}, \mathbf{w}^*)$ translates to disagreement of at most $O(\theta)$. When the target distribution is a general strongly log-concave distribution, we show a slightly weaker relationship: for any $k \in \mathbb{N}$, we can run tests requiring time $d^{\widetilde{O}(k)}$ that ensure that an angle of $\theta$ translates to disagreement of at most $O(\sqrt{k} \cdot \theta^{1-1/k})$. In the Massart noise setting, we can make $\measuredangle(\mathbf{w}, \mathbf{w}^*)$ arbitrarily small, and so obtain our $\text{opt} + \epsilon$ guarantee for any target strongly log-concave distribution in polynomial time. In the adversarial noise setting, we face a more delicate tradeoff and can only make $\measuredangle(\mathbf{w}, \mathbf{w}^*)$ as small as $\Theta(\text{opt})$. When the target distribution is Gaussian, this is enough to obtain final error $O(\text{opt}) + \epsilon$ in polynomial time. When the target distribution is a general strongly log-concave distribution, we instead obtain $\widetilde{O}(\text{opt}) + \epsilon$ in quasipolynomial time.

# 2 Preliminaries

**Notation and setup**  Throughout, the domain will be $\mathcal{X} = \mathbb{R}^d$, and labels will lie in $\mathcal{Y} = \{\pm 1\}$. The unknown joint distribution over $\mathcal{X} \times \mathcal{Y}$ that we have access to will be denoted by $D_{\mathcal{X}\mathcal{Y}}$, and its marginal on $\mathcal{X}$ will be denoted by $D_\mathcal{X}$. The target marginal on $\mathcal{X}$ will be denoted by $D^*$. We use the following convention for monomials: for a multi-index $\alpha = (\alpha_1, \ldots, \alpha_d) \in \mathbb{Z}_{\geq 0}^d$, $\mathbf{x}^\alpha$ denotes $\prod_i x_i^{\alpha_i}$, and $|\alpha| = \sum_i \alpha_i$ denotes its total degree. We use $\mathcal{C}$ to denote a concept class mapping $\mathbb{R}^d$ to $\{\pm 1\}$, which throughout this paper will be the class of halfspaces or functions of halfspaces over $\mathbb{R}^d$. We use $\text{opt}(\mathcal{C}, D_{\mathcal{X}\mathcal{Y}})$ to denote the optimal error $\inf_{f \in \mathcal{C}} \mathbb{P}_{(\mathbf{x},y) \sim D_{\mathcal{X}\mathcal{Y}}}[f(\mathbf{x}) \neq y]$, or just opt when $\mathcal{C}$ and $D_{\mathcal{X}\mathcal{Y}}$ are clear from context. We recall the definitions of the noise models we consider. In the Massart noise model, the labels satisfy $\mathbb{P}_{y \sim D_{\mathcal{X}\mathcal{Y}}|\mathbf{x}}[y \neq \text{sign}(\langle \mathbf{w}^*, \mathbf{x} \rangle) \mid \mathbf{x}] = \eta(\mathbf{x})$, where $\eta(\mathbf{x}) \leq \eta < \frac{1}{2}$ for all $\mathbf{x}$. In the adversarial label noise or agnostic model, the labels may be completely arbitrary. In both cases, the learner's goal is to produce a hypothesis with error competitive with opt.

We now formally define testable learning. The following definition is an equivalent reframing of the original definition [RV23, Def 4], folding the (label-aware) tester and learner into a single tester-learner.

**Definition 2.1** (Testable learning, [RV23])**.** Let $\mathcal{C}$ be a concept class mapping $\mathbb{R}^d$ to $\{\pm 1\}$. Let $D^*$ be a certain target marginal on $\mathbb{R}^d$. Let $\epsilon, \delta > 0$ be parameters, and let $\psi : [0, 1] \to [0, 1]$ be some function. We say $\mathcal{C}$ can be testably learned w.r.t. $D^*$ up to error $\psi(\text{opt}) + \epsilon$ with failure probability $\delta$ if there exists a tester-learner $A$ meeting the following specification. For any distribution $D_{\mathcal{X}\mathcal{Y}}$ on $\mathbb{R}^d \times \{\pm 1\}$, $A$ takes in a large sample $S$ drawn from $D_{\mathcal{X}\mathcal{Y}}$, and either rejects $S$ or accepts and produces a hypothesis $h : \mathbb{R}^d \to \{\pm 1\}$. Further, the following conditions must be met:

    (a) (Soundness.) Whenever $A$ accepts and produces a hypothesis $h$, with probability at least $1 - \delta$ (over the randomness of $S$ and $A$), $h$ must satisfy $\mathbb{P}_{(\mathbf{x},y) \sim D_{\mathcal{X}\mathcal{Y}}}[h(\mathbf{x}) \neq y] \leq \psi(\text{opt}(\mathcal{C}, D_{\mathcal{X}\mathcal{Y}})) + \epsilon$.

    (b) (Completeness.) Whenever $D_{\mathcal{X}\mathcal{Y}}$ truly has marginal $D^*$, $A$ must accept with probability at least $1 - \delta$ (over the randomness of $S$ and $A$).

# 3 Testing properties of strongly log-concave distributions

In this section we define the testers that we will need for our algorithm. All the proofs from this section can be found in Appendix B. We begin with a structural lemma that strengthens the key structural result of [GKK23], stated here as Proposition A.3. It states that even when we restrict an isotropic strongly log-concave $D^*$ to a band around the origin, moment matching suffices to fool functions of halfspaces whose weights are orthogonal to the normal of the band.

**Proposition 3.1.** *Let $D^*$ be an isotropic strongly log-concave distribution. Let $\mathbf{w} \in \mathbb{S}^{d-1}$ be any fixed direction. Let $p$ be a constant. Let $f : \mathbb{R}^d \to \mathbb{R}$ be a function of $p$ halfspaces of the form in Eq. (A.2), with the additional restriction that its weights $\mathbf{v}^i \in \mathbb{S}^{d-1}$ satisfy $\langle \mathbf{v}^i, \mathbf{w} \rangle = 0$ for all $i$. For some $\sigma \in [0,1]$, let $T$ denote the band $\{\mathbf{x} : |\langle \mathbf{w}, \mathbf{x} \rangle| \le \sigma\}$. Let $D$ be any distribution such that $D_{|T}$ matches moments of degree at most $k = \widetilde{O}(1/\tau^2)$ with $D^*_{|T}$ up to an additive slack of $d^{-\widetilde{O}(k)}$. Then $|\mathbb{E}_{D^*}[f \mid T] - \mathbb{E}_D[f \mid T]| \le \tau$.*

We now describe some of the testers that we use. First, we need a tester that ensures that the distribution is concentrated in every single direction. More formally, the tester checks that the moments of the distribution along any direction are small.

**Proposition 3.2.** *For any isotropic strongly log-concave $D^*$, there exists some constants $C_1$ and a tester $T_1$ that takes a set $S \subseteq \mathbb{R}^d \times \{\pm 1\}$, an even $k \in \mathbb{N}$, a parameter $\delta \in (0,1)$ and runs and in time $\mathrm{poly}\left(d^k, |S|, \log \frac{1}{\delta}\right)$. Let $D$ denote the uniform distribution over $S$. If $T_1$ accepts, then for any $\mathbf{v} \in \mathbb{S}^{d-1}$*

$$\mathbb{E}_{(\mathbf{x},y)\sim D}[(\langle \mathbf{v}, \mathbf{x} \rangle)^k] \le (C_1 k)^{k/2}. \tag{3.1}$$

*Moreover, if $S$ is obtained by taking at least $\left(d^k, \left(\log \frac{1}{\delta}\right)^k\right)^{C_1}$ i.i.d. samples from a distribution whose $\mathbb{R}^d$-marginal is $D^*$, the test $T_1$ passes with probability at least $1 - \delta$.*

Secondly, we will use a tester that makes sure the distribution is not concentrated too close to a specific hyperplane. This is one of the properties we will need to use in order to employ the localization technique of [ABL17].

**Proposition 3.3.** *For any isotropic strongly log-concave $D^*$, there exist some constants $C_2, C_3$ and a tester $T_2$ that takes a set $S \subseteq \mathbb{R}^d \times \{\pm 1\}$ a vector $\mathbf{w} \in \mathbb{S}^{d-1}$, parameters $\sigma, \delta \in (0,1)$ and runs in time $\mathrm{poly}\left(d, |S|, \log \frac{1}{\delta}\right)$. Let $D$ denote the uniform distribution over $S$. If $T_2$ accepts, then*

$$\mathbb{P}_{(\mathbf{x},y)\sim D}[|\langle \mathbf{w}, \mathbf{x} \rangle| \le \sigma] \in (C_2\sigma, C_3\sigma). \tag{3.2}$$

*Moreover, if $S$ is obtained by taking at least $\frac{100}{K_1\sigma^2} \log \left(\frac{1}{\delta}\right)$ i.i.d. samples from a distribution whose $\mathbb{R}^d$-marginal is $D^*$, the test $T_2$ passes with probability at least $1 - \delta$.*

Finally, in order to use the localization idea of [ABL17] in a manner similar to [DKTZ20b], we need to make sure that the distribution is well-behaved also within a band around to a certain hyperplane. The main property of the distribution that we establish is that functions of constantly many halfspaces have expectations very close to what they would be under our distributional assumption. As we show later in this work, having the aforementioned property allows us to derive many other properties that strongly log-concave distributions have, including many of the key properties that make the localization technique successful.

**Proposition 3.4.** *For any isotropic strongly log-concave $D^*$ and a constant $C_4$, there exists a constant $C_5$ and a tester $T_3$ that takes a set $S \subseteq \mathbb{R}^d \times \{\pm 1\}$ a vector $\mathbf{w} \in \mathbb{S}^{d-1}$, parameters $\sigma, \tau \, \delta \in (0,1)$ and runs in time $\mathrm{poly}\left(d^{\tilde{O}\left(\frac{1}{\tau^2}\right)}, \frac{1}{\sigma}, |S|, \log \frac{1}{\delta}\right)$. Let $D$ denote the uniform distribution over $S$, let $T$ denote the band $\{\mathbf{x} : |\langle \mathbf{w}, \mathbf{x} \rangle| \le \sigma\}$ and let $\mathcal{F}_{\mathbf{w}}$ denote the set $\{\pm 1\}$-valued functions of $C_4$ halfspaces whose weight vectors are orthogonal to $\mathbf{w}$. If $T_3$ accepts, then*

$$\max_{f \in \mathcal{F}_{\mathbf{w}}} \left| \mathbb{E}_{\mathbf{x}\sim D^*}[f(\mathbf{x}) \mid \mathbf{x} \in T] - \mathbb{E}_{(\mathbf{x},y)\sim D}[f(\mathbf{x}) \mid \mathbf{x} \in T] \right| \le \tau, \tag{3.3}$$

$$\max_{\mathbf{v} \in \mathbb{S}^{d-1}: \langle \mathbf{v},\mathbf{w} \rangle = 0} \left| \mathbb{E}_{\mathbf{x}\sim D^*}[(\langle \mathbf{v}, \mathbf{x} \rangle)^2 \mid \mathbf{x} \in T] - \mathbb{E}_{(\mathbf{x},y)\sim D}[(\langle \mathbf{v}, \mathbf{x} \rangle)^2 \mid \mathbf{x} \in T] \right| \le \tau. \tag{3.4}$$

Moreover, if $S$ is obtained by taking at least $\left( \frac{1}{\tau} \cdot \frac{1}{\sigma} \cdot d^{\frac{1}{\tau^2} \log^{C_5}\left(\frac{1}{\tau}\right)} \cdot \left(\log \frac{1}{\delta}\right)^{\frac{1}{\tau^2} \log^{C_5}\left(\frac{1}{\tau}\right)} \right)^{C_5}$ i.i.d. samples from a distribution whose $\mathbb{R}^d$-marginal is $D^*$, the test $T_3$ passes with probability at least $1 - \delta$.

# 4 Testably learning halfspaces with Massart noise

In this section we prove that we can testably learn halfspaces with Massart noise with respect to isotropic strongly log-concave distributions (see Definition A.1).

**Theorem 4.1** (Tester-Learner for Halfspaces with Massart Noise). *Let $D_{\mathcal{XY}}$ be a distribution over $\mathbb{R}^d \times \{\pm 1\}$ and let $D^*$ be an isotropic strongly log-concave distribution over $\mathbb{R}^d$. Let $\mathcal{C}$ be the class of origin centered halfspaces in $\mathbb{R}^d$. Then, for any $\eta < 1/2$, $\epsilon > 0$ and $\delta \in (0, 1)$, there exists an algorithm (Algorithm 1) that testably learns $\mathcal{C}$ w.r.t. $D^*$ up to excess error $\epsilon$ and error probability at most $\delta$ in the Massart noise model with rate at most $\eta$, using time and a number of samples from $D_{\mathcal{XY}}$ that are polynomial in $d, 1/\epsilon, \frac{1}{1-2\eta}$ and $\log(1/\delta)$.*

---

**Algorithm 1:** Tester-learner for halfspaces

**Input:** Training sets $S_1, S_2$, parameters $\sigma, \delta, \alpha$
**Output:** A near-optimal weight vector $\mathbf{w}$, or rejection
Run PSGD on the empirical loss $\mathcal{L}_\sigma$ over $S_1$ to get a list $L$ of candidate vectors.
Test whether $L$ contains an $\alpha$-approximate stationary point $\mathbf{w}$ of the empirical loss $\mathcal{L}_\sigma$ over $S_2$.
 Reject if no such $\mathbf{w}$ exists.
**for** *each candidate $\mathbf{w}'$ in $\{\mathbf{w}, -\mathbf{w}\}$* **do**
  Let $B'_{\mathbf{w}}(\sigma)$ denote the band $\{\mathbf{x} : |\langle \mathbf{w}', \mathbf{x}\rangle| \leq \sigma\}$. Let $\mathcal{F}'_{\mathbf{w}}$ denote the class of functions of at
    most two halfspaces with weights orthogonal to $\mathbf{w}'$.
  Let $\delta' = \Theta(\delta)$.
  Run $T_1(S_2, k = 2, \delta)$ to verify that the empirical marginal is approximately isotropic. Reject
    if $T_1$ rejects.
  Run $T_2(S_2, \mathbf{w}', \sigma, \delta')$ to verify that $\mathbb{P}_S[B'_{\mathbf{w}}] = \Theta(\sigma)$. Reject if $T_2$ rejects.
  Run $T_3(S_2, \mathbf{w}', \sigma = \sigma/6, \tau, \delta')$ and $T_3(S, \mathbf{w}', \sigma = \sigma/2, \tau, \delta')$ for a suitable constant $\tau$ to
    verify that the empirical distribution conditioned on $B'_{\mathbf{w}}(\sigma/6)$ and $B'_{\mathbf{w}}(\sigma/2)$ fools $\mathcal{F}'_{\mathbf{w}}$ up to
    $\tau$. Reject if $T_3$ rejects.
  Estimate the empirical error of $\mathbf{w}'$ on $S$.
If all tests have accepted, output $\mathbf{w}' \in \{\mathbf{w}, -\mathbf{w}\}$ with the best empirical error.

---

To show our result, we revisit the approach of [DKTZ20a] for learning halfspaces with Massart noise under well-behaved distributions. Their result is based on the idea of minimizing a surrogate loss that is non convex, but whose stationary points correspond to halfspaces with low error. They also require that their surrogate loss is sufficiently smooth, so that one can find a stationary point efficiently. While the distributional assumptions that are used to demonstrate that stationary points of the surrogate loss can be discovered efficiently are mild, the main technical lemma, which demostrates that any stationary point suffices, requires assumptions that are not necessarily testable. We establish a label-dependent approach for testing, making use of tests that are applied during the course of our algorithm.

We consider a slightly different surrogate loss than the one used in [DKTZ20a]. In particular, for $\sigma > 0$, we let

$$\mathcal{L}_\sigma(\mathbf{w}) = \mathbb{E}_{(\mathbf{x},y) \sim D_{\mathcal{XY}}}\left[\ell_\sigma\left(-y\frac{\langle \mathbf{w}, \mathbf{x}\rangle}{\|\mathbf{w}\|_2}\right)\right], \tag{4.1}$$

where $\ell_\sigma : \mathbb{R} \to [0, 1]$ is a smooth approximation to the ramp function with the properties described in Proposition C.1 (see Appendix C), obtained using a piecewise polynomial of degree 3. Unlike the standard logistic function, our loss function has derivative exactly 0 away from the origin (for $|t| > \sigma/2$). This makes the analysis of the gradient of $\mathcal{L}_\sigma$ easier, since the contribution from points lying outside a certain band is exactly 0.

The smoothness allows us to run PSGD to obtain stationary points efficiently, and we now state the convergence lemma we need.

**Proposition 4.2** (PSGD Convergence, Lemmas 4.2 and B.2 in [DKTZ20a]). *Let $\mathcal{L}_\sigma$ be as in Equation (4.1) with $\sigma \in (0,1]$, $\ell_\sigma$ as described in Proposition C.1 and $D_{\mathcal{X}\mathcal{Y}}$ such that the marginal $D_{\mathcal{X}}$ on $\mathbb{R}^d$ satisfies Property (3.1) for $k = 2$. Then, for any $\epsilon > 0$ and $\delta \in (0,1)$, there is an algorithm whose time and sample complexity is $O(\frac{d}{\sigma^4} + \frac{\log(1/\delta)}{\epsilon^4 \sigma^4})$, which, having access to samples from $D_{\mathcal{X}\mathcal{Y}}$, outputs a list $L$ of vectors $\mathbf{w} \in \mathbb{S}^{d-1}$ with $|L| = O(\frac{d}{\sigma^4} + \frac{\log(1/\delta)}{\epsilon^4 \sigma^4})$ so that there exists $\mathbf{w} \in L$ with*

$$\|\nabla_{\mathbf{w}} \mathcal{L}_\sigma(\mathbf{w})\|_2 \leq \epsilon \,, \text{ with probability at least } 1 - \delta \,.$$

*In particular, the algorithm performs Stochastic Gradient Descent on $\mathcal{L}_\sigma$ Projected on $\mathbb{S}^{d-1}$ (PSGD).*

It now suffices to show that, upon performing PSGD on $\mathcal{L}_\sigma$, for some appropriate choice of $\sigma$, we acquire a list of vectors that testably contain a vector which is approximately optimal. We first prove the following lemma, whose distributional assumptions are relaxed compared to the corresponding structural Lemma 3.2 of [DKTZ20a]. In particular, instead of requiring the marginal distribution to be "well-behaved", we assume that the quantities of interest (for the purposes of our proof) have expected values under the true marginal distribution that are close, up to multiplicative factors, to their expected values under some "well-behaved" (in fact, strongly log-concave) distribution. While some of the quantities of interest have values that are miniscule and estimating them up to multiplicative factors could be too costly, it turns out that the source of their vanishing scaling can be completely attributed to factors of the form $\mathbb{P}[|\langle \mathbf{w}, \mathbf{x} \rangle| \leq \sigma]$ (where $\sigma$ is small), which, due to standard concentration arguments, can be approximated up to multiplicative factors, given $\mathbf{w} \in \mathbb{S}^{d-1}$ and $\sigma > 0$ (see Proposition 3.3). As a result, we may estimate the remaining factors up to sufficiently small additive constants (see Proposition 3.4) to get multiplicative overall closeness to the "well behaved" baseline. We defer the proof of the following Lemma to Appendix C.1.

**Lemma 4.3.** *Let $\mathcal{L}_\sigma$ be as in Equation (4.1) with $\sigma \in (0,1]$, $\ell_\sigma$ as described in Proposition C.1, let $\mathbf{w} \in \mathbb{S}^{d-1}$ and consider $D_{\mathcal{X}\mathcal{Y}}$ such that the marginal $D_{\mathcal{X}}$ on $\mathbb{R}^d$ satisfies Properties (3.2) and (3.3) for $C_4 = 2$ and accuracy $\tau$. Let $\mathbf{w}^* \in \mathbb{S}^{d-1}$ define an optimum halfspace and let $\eta < 1/2$ be an upper bound on the rate of the Massart noise. Then, there are constants $c_1, c_2, c_3 > 0$ such that if $\|\nabla_{\mathbf{w}} \mathcal{L}_\sigma(\mathbf{w})\|_2 < c_1(1 - 2\eta)$ and $\tau \leq c_2$, then*

$$\measuredangle(\mathbf{w}, \mathbf{w}^*) \leq \frac{c_3}{1 - 2\eta} \cdot \sigma \quad or \quad \measuredangle(-\mathbf{w}, \mathbf{w}^*) \leq \frac{c_3}{1 - 2\eta} \cdot \sigma$$

Combining Proposition 4.2 and Lemma 4.3, we get that for any choice of the parameter $\sigma \in (0,1]$, by running PSGD on $\mathcal{L}_\sigma$, we can construct a list of vectors of polynomial size (in all relevant parameters) that testably contains a vector that is close to the optimum weight vector. In order to link the zero-one loss to the angular similarity between a weight vector and the optimum vector, we use the following Proposition (for the proof, see Appendix C.2).

**Proposition 4.4.** *Let $D_{\mathcal{X}\mathcal{Y}}$ be a distribution over $\mathbb{R}^d \times \{\pm 1\}$, $\mathbf{w}^* \in \arg\min_{\mathbf{w} \in \mathbb{S}^{d-1}} \mathbb{P}_{D_{\mathcal{X}\mathcal{Y}}}[y \neq \operatorname{sign}(\langle \mathbf{w}, \mathbf{x} \rangle)]$ and $\mathbf{w} \in \mathbb{S}^{d-1}$. Then, for any $\theta \geq \measuredangle(\mathbf{w}, \mathbf{w}^*)$, $\theta \in [0, \pi/4]$, if the marginal $D_{\mathcal{X}}$ on $\mathbb{R}^d$ satisfies Property (3.1) for $C_1 > 0$ and some even $k \in \mathbb{N}$ and Property (3.2) with $\sigma$ set to $(C_1 k)^{\frac{k}{2(k+1)}} \cdot (\tan \theta)^{\frac{k}{k+1}}$, then, there exists a constant $c > 0$ such that the following is true.*

$$\mathbb{P}_{D_{\mathcal{X}\mathcal{Y}}}[y \neq \operatorname{sign}(\langle \mathbf{w}, \mathbf{x} \rangle)] \leq \operatorname{opt} + c \cdot k^{1/2} \cdot \theta^{1 - \frac{1}{k+1}} \,.$$

We are now ready to prove Theorem 4.1.

*Proof of Theorem 4.1.* Throughout the proof we consider $\delta'$ to be a sufficiently small polynomial in all the relevant parameters. Each of the failure events will have probability at least $\delta'$ and their number will be polynomial in all the relevant parameters, so by the union bound, we may pick $\delta'$ so that the probability of failure is at most $\delta$.

The algorithm we run is Algorithm 1, with appropriate selection of parameters and given samples $S_1, S_2$, each of which are sufficiently large sets of independent samples from the true unknown distribution $D_{\mathcal{X}\mathcal{Y}}$. For some $\sigma \in (0,1]$ to be defined later, we run PSGD on the empirical loss $\mathcal{L}_\sigma$ over $S_1$ as described in Proposition 4.2 with $\epsilon = c_1(1 - 2\eta)\sigma/4$, where $c_1$ is given by Lemma 4.3. By Proposition 4.2, we get a list $L$ of vectors $\mathbf{w} \in \mathbb{S}^{d-1}$ with $|L| = \operatorname{poly}(d, 1/\sigma)$ such that there exists $\mathbf{w} \in L$ with $\|\nabla_{\mathbf{w}} \mathcal{L}_\sigma(\mathbf{w})\|_2 < \frac{1}{2}c_1(1 - 2\eta)$ under the true distribution, if the marginal is isotropic.

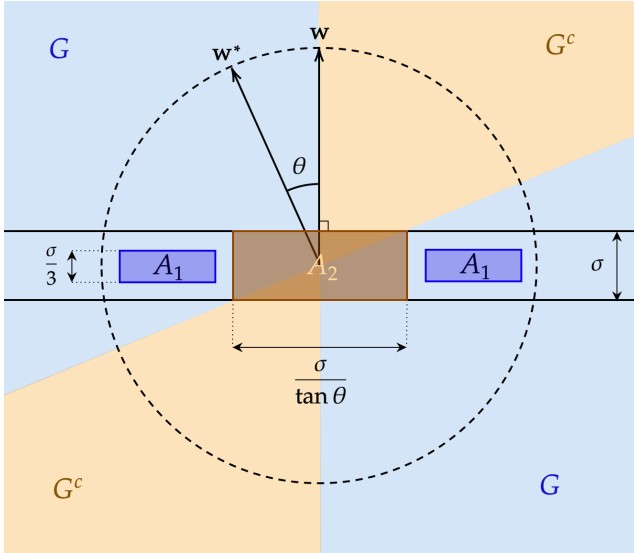

Figure 1: Critical regions in the proofs of main structural lemmas (Lemmas 4.3, 5.2). We analyze the contributions of the regions labeled $A_1, A_2$ to the quantities $A_1, A_2$ in the proofs. Specifically, the regions $A_1$ (which have height $\sigma/3$ so that the value of $\ell'_\sigma(\mathbf{x_w})$ for any $\mathbf{x}$ in these regions is exactly $1/\sigma$, by Proposition C.1) form a subset of the region $\mathcal{G}$, and their probability mass under $D_\mathcal{X}$ is (up to a multiplicative factor) a lower bound on the quantity $A_1$ (see Eq (C.3)). Similarly, the region $A_2$ is a subset of the intersection of $\mathcal{G}^c$ with the band of height $\sigma$, and has probability mass that is (up to a multiplicative factor) an upper bound on the quantity $A_2$ (see Eq (C.4)).

315 Having acquired the list $L$ using sample $S_1$, we use the independent samples in $S_2$ to test whether
316 $L$ contains an approximately stationary point of the empirical loss on $S_2$. If this is not the case,
317 then we may safely reject: for large enough $|S_1|$, if the distribution is indeed isotropic strongly
318 logconcave, there is an approximate stationary of the population loss in $L$ and if $|S_2|$ is large enough,
319 the gradient of the empirical loss on $S_2$ will be close to the gradient of the population loss on each of
320 the elements of $L$, due to appropriate concentration bounds for log-concave distributions as well as
321 the fact that the elements of $L$ are independent from $S_2$. For the following, let $\mathbf{w}$ be a point such that
322 $\|\nabla_\mathbf{w}\mathcal{L}_\sigma(\mathbf{w})\|_2 < c_1(1-2\eta)$ under the empirical distribution over $S_2$

323 In Lemma 4.3 and Proposition 4.4 we have identified certain properties of the marginal distribution
324 that are sufficient for our purposes, given that $L$ contains an approximately stationary point of the
325 empirical (surrogate) loss on $S_2$. Our testers $T_1, T_2, T_3$ verify that these properties hold for the
326 empirical marginal over our sample $S_2$, and it will be convenient to analyze the optimality of our
327 algorithm purely over $S_2$. In particular, we will need to require that $|S_2|$ is sufficiently large, so
328 that when the true marginal is indeed the target $D^*$, our testers succeed with high probability (for
329 the corresponding sample complexity, see Propositions 3.2, 3.3 and 3.4). Moreover, by standard
330 generalization theory, since the VC dimension of halfspaces is only $O(d)$ and for us $|S_2|$ is a large
331 poly$(d, 1/\epsilon)$, both the error of our final output and the optimal error over $S_2$ will be close to that over
332 $D_{\mathcal{X}\mathcal{Y}}$. So in what follows, we will abuse notation and refer to the uniform distribution over $S_2$ as
333 $D_{\mathcal{X}\mathcal{Y}}$ and the optimal error over $S_2$ simply as opt.

334 We proceed with some basic tests. Throughout the rest of the algorithm, whenever a tester fails,
335 we reject, otherwise we proceed. First, we run testers $T_2$ with inputs $(\mathbf{w}, \sigma/2, \delta')$ and $(\mathbf{w}, \sigma/6, \delta')$
336 (Proposition 3.3) and $T_3$ with inputs $(\mathbf{w}, \sigma/2, c_2, \delta')$ and with $(\mathbf{w}, \sigma/6, c_2, \delta')$ (Proposition 3.4, $c_2$
337 as defined in Lemma 4.3). This ensures that for the approximate stationary point $\mathbf{w}$ of the $\mathcal{L}_\sigma$, the
338 probability within the band $B_\mathbf{w}(\sigma/2) = \{\mathbf{x} : |\langle\mathbf{w}, \mathbf{x}\rangle| \leq \sigma/2\}$ is $\Theta(\sigma)$ (and similarly for $B_\mathbf{w}(\sigma/6)$)
339 and moreover that our marginal conditioned on each of the bands fools (up to an additive constant)
340 functions of halfspaces with weights orthogonal to $\mathbf{w}$. As a result, we may apply Lemma 4.3 to
341 $\mathbf{w}$ and form a list of 2 vectors $\{\mathbf{w}, -\mathbf{w}\}$ which contains some $\mathbf{w}'$ with $\measuredangle(\mathbf{w}', \mathbf{w}^*) \leq c_2\sigma/(1-2\eta)$
342 (where $c_3$ is as defined in Lemma 4.3).

We run $T_1$ (Proposition 3.2) with $k = 2$ to verify that the marginals are approximately isotropic and
we use $T_2$ once again, with appropriate parameters for each $\mathbf{w}$ and its negation, to apply Proposition
4.4 and get that $\{\mathbf{w}, -\mathbf{w}\}$ contains a vector $\mathbf{w}'$ with

$$\mathbb{P}_{D_{\mathcal{X}\mathcal{Y}}}[y \neq \mathrm{sign}(\langle \mathbf{w}', \mathbf{x} \rangle)] \leq \mathsf{opt} + c \cdot \theta^{2/3},$$

where $\measuredangle(\mathbf{w}', \mathbf{w}^*) \leq \theta := c_2 \sigma / \sqrt{1 - 2\eta}$. By picking $\sigma = \Theta(\epsilon^{3/2}(1 - 2\eta))$, we get

$$\mathbb{P}_{D_{\mathcal{X}\mathcal{Y}}}[y \neq \mathrm{sign}(\langle \mathbf{w}', \mathbf{x} \rangle)] \leq \mathsf{opt} + \epsilon.$$

However, we do not know which of the weight vectors in $\{\mathbf{w}, -\mathbf{w}\}$ is the one guaranteed to achieve
small error. In order to discover this vector, we estimate the probability of error of each of the
corresponding halfspaces (which can be done efficiently, due to Hoeffding's bound) and pick the one
with the smallest error. This final step does not require any distributional assumptions and we do not
need to perform any further tests. $\qquad\square$

# 5 Testably learning halfspaces in the agnostic setting

In this section, we provide our result on efficiently and testably learning halfspaces in the agnostic
setting with respect to isotropic strongly log-concave target marginals. We defer the proofs to
Appendix D. The algorithm we use is once more Algorithm 1, but we call it multiple times for
different choices of the parameter $\sigma$, reject if any call rejects and output the vector that achieved
the minimum empirical error overall, otherwise. Also, the tester $T_1$ is called for a general $k$ (not
necessarily $k = 2$).

**Theorem 5.1** (Efficient Tester-Learner for Halfspaces in the Agnostic Setting)**.** *Let $D_{\mathcal{X}\mathcal{Y}}$ be a
distribution over $\mathbb{R}^d \times \{\pm 1\}$ and let $D^*$ be a strongly log-concave distribution over $\mathbb{R}^d$ (Definition
A.1). Let $\mathcal{C}$ be the class of origin centered halfspaces in $\mathbb{R}^d$. Then, for any even $k \in \mathbb{N}$, any $\epsilon > 0$
and $\delta \in (0, 1)$, there exists an algorithm that agnostically testably learns $\mathcal{C}$ w.r.t. $D^*$ up to error
$O(k^{1/2} \cdot \mathsf{opt}^{1 - \frac{1}{k+1}}) + \epsilon$, where $\mathsf{opt} = \min_{\mathbf{w} \in \mathbb{S}^{d-1}} \mathbb{P}_{D_{\mathcal{X}\mathcal{Y}}}[y \neq \mathrm{sign}(\langle \mathbf{w}, \mathbf{x} \rangle)]$, and error probability
at most $\delta$, using time and a number of samples from $D_{\mathcal{X}\mathcal{Y}}$ that are polynomial in $d^{\tilde{O}(k)}, (1/\epsilon)^{\tilde{O}(k)}$
and $(\log(1/\delta))^{O(k)}$.*

*In particular, by picking some appropriate $k \leq \log^2 d$, we obtain error $\tilde{O}(\mathsf{opt}) + \epsilon$ in quasipolynomial
time and sample complexity, i.e. $\mathrm{poly}(2^{\mathrm{polylog}\, d}, (\frac{1}{\epsilon})^{\mathrm{polylog}\, d})$.*

To prove Theorem 5.1, we may follow a similar approach as the one we used for the case of Massart
noise. However, in this case, the main structural lemma regarding the quality of the stationary points
involves an additional requirement about the parameter $\sigma$. In particular, $\sigma$ cannot be arbitrarily small
with respect to the error of the optimum halfspace, because, in this case, there is no upper bound
on the amount of noise that any specific point $\mathbf{x}$ might be associated with. As a result, picking $\sigma$
to be arbitrarily small would imply that our algorithm only considers points that lie within a region
that has arbitrarily small probability and can hence be completely corrupted with the adversarial
opt budget. On the other hand, the polynomial slackness that the testability requirement introduces
(through Proposition 4.4) between the error we achieve and the angular distance guarantee we can get
via finding a stationary point of $\mathcal{L}_\sigma$ (which is now coupled with opt), appears to the exponent of the
guarantee we achieve in Theorem 5.1.

**Lemma 5.2.** *Let $\mathcal{L}_\sigma$ be as in Equation (4.1) with $\sigma \in (0, 1]$, $\ell_\sigma$ as described in Proposition C.1, let
$\mathbf{w} \in \mathbb{S}^{d-1}$ and consider $D_{\mathcal{X}\mathcal{Y}}$ such that the marginal $D_{\mathcal{X}}$ on $\mathbb{R}^d$ satisfies Properties (3.2), (3.3) and
(3.4) for $\mathbf{w}$ with $C_4 = 2$ and accuracy parameter $\tau$. Let opt be the minimum error achieved by some
origin centered halfspace and let $\mathbf{w}^* \in \mathbb{S}^{d-1}$ be a corresponding vector. Then, there are constants
$c_1, c_2, c_3, c_4 > 0$ such that if $\mathsf{opt} \leq c_1 \sigma$, $\|\nabla_{\mathbf{w}} \mathcal{L}_\sigma(\mathbf{w})\|_2 < c_2$, and $\tau \leq c_3$ then*

$$\measuredangle(\mathbf{w}, \mathbf{w}^*) \leq c_4 \sigma \quad or \quad \measuredangle(-\mathbf{w}, \mathbf{w}^*) \leq c_4 \sigma.$$

We obtain our main result for Gaussian target marginals by refining Proposition 4.4 for the specific
case when the target marginal distribution $D^*$ is the standard multivariate Gaussian distribution. The
algorithm for the Gaussian case is similar to the one of Theorem 5.1, but it runs different tests for the
improved version (see Proposition D.1) of Proposition 4.4.

**Theorem 5.3.** *In Theorem 5.1, if $D^*$ is the standard Gaussian in $d$ dimensions, we obtain error
$O(\mathsf{opt}) + \epsilon$ in polynomial time and sample complexity, i.e. $\mathrm{poly}(d, 1/\epsilon, \log(1/\delta))$.*

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
