# A   Strongly log-concave distributions

473  We also formally define the class of *strongly log-concave* distributions, which is the class that our
474  target marginal $D^*$ is allowed to belong to, and collect some useful properties of such distributions.
475  We will state the definition for isotropic $D^*$ (i.e. with mean 0 and covariance $I$) for simplicity.

476  **Definition A.1** (Strongly log-concave distribution, see e.g. [SW14, Def 2.8])**.** We say an isotropic
477  distribution $D^*$ on $\mathbb{R}^d$ is strongly log-concave if the logarithm of its density $q$ is a strongly concave
478  function. Equivalently, $q$ can be written as

$$q(\mathbf{x}) = r(\mathbf{x})\gamma_{\kappa^2 I}(\mathbf{x}) \tag{A.1}$$

479  for some log-concave function $r$ and some constant $\kappa > 0$, where $\gamma_{\kappa^2 I}$ denotes the density of the
480  spherical Gaussian $\mathcal{N}(0, \kappa^2 I)$.

481  **Proposition A.2** (see e.g. [SW14])**.** *Let $D^*$ be an isotropic strongly log-concave distribution on $\mathbb{R}^d$*
482  *with density $q$.*

483      *(a) Any orthogonal projection of $D^*$ onto a subspace is also strongly log-concave.*

484      *(b) There exist constants $U, R$ such that $q(\mathbf{x}) \leq U$ for all $\mathbf{x}$, and $q(x) \geq 1/U$ for all $\|\mathbf{x}\| \leq R$.*

485      *(c) There exist constants $U'$ and $\kappa$ such that $q(\mathbf{x}) \leq U'\gamma_{\kappa^2 I}(\mathbf{x})$ for all $\mathbf{x}$.*

486      *(d) There exist constants $K_1, K_2$ such that for any $\sigma \in [0, 1]$ and any $\mathbf{v} \in \mathbb{S}^{d-1}$, $\mathbb{P}[|\langle \mathbf{v}, \mathbf{x}\rangle| \leq$*
487          *$\sigma] \in (K_1\sigma, K_2\sigma)$.*

488      *(e) There exists a constant $K_3$ such that for any $k \in \mathbb{N}$, $\mathbb{E}[|\langle \mathbf{v}, \mathbf{x}\rangle|^k] \leq (K_3 k)^{k/2}$.*

489      *(f) Let $\alpha = (\alpha_1, \ldots, \alpha_d) \in \mathbb{Z}_{\geq 0}^d$ be a multi-index with total degree $|\alpha| = \sum_i \alpha_i = k$, and let*
490          *$\mathbf{x}^\alpha = \prod_i x_i^{\alpha_i}$. There exists a constant $K_4$ such that for any such $\alpha$, $\mathbb{E}[|\mathbf{x}^\alpha|] \leq (K_4 k)^{k/2}$.*

491  For (a), see e.g. [SW14, Thm 3.7]. The other properties follow readily from Eq. (A.1), which allows
492  us to treat the density as subgaussian.

493  A key structural fact that we will need about strongly log-concave distributions is that approximately
494  matching moments of degree at most $\widetilde{O}(1/\tau^2)$ with such a $D^*$ is sufficient to fool any function of a
495  constant number of halfspaces up to an additive $\tau$.

496  **Proposition A.3** (Variant of [GKK23, Thm 5.6])**.** *Let $p$ be a fixed constant, and let $\mathcal{F}$ be the class of*
497  *all functions of $p$ halfspaces mapping $\mathbb{R}^d$ to $\{\pm 1\}$ of the form*

$$f(\mathbf{x}) = g\left(\text{sign}(\langle \mathbf{v}^1, \mathbf{x}\rangle + \theta_1), \ldots, \text{sign}(\langle \mathbf{v}^p, \mathbf{x}\rangle + \theta_p)\right) \tag{A.2}$$

498  *for some $g : \{\pm 1\}^p \to \{\pm 1\}$ and weights $\mathbf{v}^i \in \mathbb{S}^{d-1}$. Let $D^*$ be any target marginal such that*
499  *for every $i$, the projection $\langle \mathbf{v}^i, \mathbf{x}\rangle$ has subgaussian tails and is anticoncentrated: (a) $\mathbb{P}[|\langle \mathbf{v}^i, \mathbf{x}\rangle| >$*
500  *$t] \leq \exp(-\Theta(t^2))$, and (b) for any interval $[a, b]$, $\mathbb{P}[\langle \mathbf{v}^i, \mathbf{x}\rangle \in [a, b]] \leq \Theta(|b - a|)$. Let $D$ be any*
501  *distribution such that for all monomials $\mathbf{x}^\alpha = \prod_i x_i^{\alpha_i}$ of total degree $|\alpha| = \sum_i \alpha_i \leq k$,*

$$\left|\mathop{\mathbb{E}}_{D^*}[\mathbf{x}^\alpha] - \mathop{\mathbb{E}}_{D}[\mathbf{x}^\alpha]\right| \leq \left(\frac{c|\alpha|}{d\sqrt{k}}\right)^{|\alpha|}$$

502  *for some sufficiently small constant $c$ (in particular, it suffices to have $d^{-\widetilde{O}(k)}$ moment closeness for*
503  *every $\alpha$). Then*

$$\max_{f \in \mathcal{F}} \left|\mathop{\mathbb{E}}_{D^*}[f] - \mathop{\mathbb{E}}_{D}[f]\right| \leq \widetilde{O}\left(\frac{1}{\sqrt{k}}\right).$$

504  Note that this is a variant of the original statement of [GKK23, Thm 5.6], which requires that the 1D
505  projection of $D^*$ along *any* direction satisfy suitable concentration and anticoncentration. Indeed, an
506  inspection of their proof reveals that it suffices to verify these properties for projections only along
507  the directions $\{\mathbf{v}^i\}_{i \in [p]}$ as opposed to all directions. This is because to fool a function $f$ of the form
508  above, their proof only analyzes the projected distribution $(\langle \mathbf{v}^1, \mathbf{x}\rangle, \ldots, \langle \mathbf{v}^p, \mathbf{x}\rangle)$ on $\mathbb{R}^p$, and requires
509  only concentration and anticoncentration for each individual projection $\langle \mathbf{v}^i, \mathbf{x}\rangle$.

# B  Proofs for Section 3

## B.1  Proof of Proposition 3.1

Our plan is to apply Proposition A.3. To do so, we must verify that $D^*_{|T}$ satisfies the assumptions required. In particular, it suffices to verify that the 1D projection along any direction orthogonal to $\mathbf{w}$ has subgaussian tails and is anticoncentrated. Let $\mathbf{v} \in \mathbb{S}^{d-1}$ be any direction that is orthogonal to $\mathbf{w}$. By Proposition A.2(d), we may assume that $\mathbb{P}_{D^*}[T] \geq \Omega(\sigma)$.

To verify subgaussian tails, we must show that for any $t$, $\mathbb{P}_{D^*_{|T}}[|\langle \mathbf{v}, \mathbf{x} \rangle| > t] \leq \exp(-Ct^2)$ for some constant $C$. The main fact we use is Proposition A.2(c), i.e. that any strongly log-concave density is pointwise upper bounded by a Gaussian density times a constant. Write

$$\mathbb{P}_{D^*_{|T}}[|\langle \mathbf{v}, \mathbf{x} \rangle| > t] = \frac{\mathbb{P}_{D^*}[\langle \mathbf{v}, \mathbf{x} \rangle > t \text{ and } \langle \mathbf{w}, \mathbf{x} \rangle \in [-\sigma, \sigma]]}{\mathbb{P}_{D^*}[\langle \mathbf{w}, \mathbf{x} \rangle \in [-\sigma, \sigma]]}.$$

The claim now follows from the fact that the numerator is upper bounded by a constant times the corresponding probability under a Gaussian density, which is at most $O(\exp(-C't^2)\sigma)$ for some constant $C'$, and that the denominator is $\Omega(\sigma)$.

To check anticoncentration, for any interval $[a, b]$, write

$$\mathbb{P}_{D^*_{|T}}[\langle \mathbf{v}, \mathbf{x} \rangle \in [a, b]] = \frac{\mathbb{P}_{D^*}[\langle \mathbf{v}, \mathbf{x} \rangle \in [a, b] \text{ and } \langle \mathbf{w}, \mathbf{x} \rangle \in [-\sigma, \sigma]]}{\mathbb{P}_{D^*}[\langle \mathbf{w}, \mathbf{x} \rangle \in [-\sigma, \sigma]]}.$$

After projecting onto $\mathrm{span}(\mathbf{v}, \mathbf{w})$ (an operation that preserves logconcavity), the numerator is the probability mass under a rectangle with side lengths $|b - a|$ and $2\sigma$, which is at most $O(\sigma|b - a|)$ as by Proposition A.2(b) the density is pointwise upper bounded by a constant. The claim follows since the denominator is $\Omega(\sigma)$.

Now we are ready to apply Proposition A.3. We see that if $D_{|T}$ matches moments of degree at most $k$ with $D^*_{|T}$ up to an additive slack of $d^{-O(k)}$, then $|\mathbb{E}_{D^*}[f \mid T] - \mathbb{E}_D[f \mid T]| \leq \widetilde{O}(1/\sqrt{k})$. Rewriting in terms of $\tau$ gives the theorem.

## B.2  Proof of Proposition 3.2

The tester $T_1$ does the following:

    1. For all $\alpha \in \mathbb{Z}^d_{\geq 0}$ with $|\alpha| = k$:

        (a) Compute the corresponding moment $\mathbb{E}_{(\mathbf{x}, y) \sim D} \mathbf{x}^\alpha := \frac{1}{|S|} \sum_{\mathbf{x} \in S} \mathbf{x}^\alpha$.

        (b) If $\left| \mathbb{E}_{(\mathbf{x}, y) \sim D}[\mathbf{x}^\alpha] - \mathbb{E}_{\mathbf{x} \sim D^*}[\mathbf{x}^\alpha] \right| > \frac{1}{d^k}$ then reject.

    2. If all the checks above passed, accept.

First, we claim that for some absolute constant $C_1$, if the tester above accepts, we have $\mathbb{E}_{(\mathbf{x}, y) \sim D}[(\langle \mathbf{v}, \mathbf{x} \rangle)^k] \leq (C_1 k)^{k/2}$ for any $\mathbf{v} \in \mathbb{S}^{d-1}$. To show this, we first recall that by Proposition A.2(e) it is the case that $\mathbb{E}_{(\mathbf{x}, y) \sim D^*}[(\langle \mathbf{v}, \mathbf{x} \rangle)^k] \leq (K_3 k)^{k/2}$. But we have

$$\left| \mathbb{E}_{(\mathbf{x}, y) \sim D}[(\langle \mathbf{v}, \mathbf{x} \rangle)^k] - \mathbb{E}_{(\mathbf{x}, y) \sim D^*}[(\langle \mathbf{v}, \mathbf{x} \rangle)^k] \right| \leq \sum_{\alpha : |\alpha| = k} \left| \mathbb{E}_{(\mathbf{x}, y) \sim D}[\mathbf{x}^\alpha] - \mathbb{E}_{\mathbf{x} \sim D^*}[\mathbf{x}^\alpha] \right|$$

$$\leq d^k \cdot \max_{\alpha : |\alpha| = k} \left| \mathbb{E}_{(\mathbf{x}, y) \sim D}[\mathbf{x}^\alpha] - \mathbb{E}_{\mathbf{x} \sim D^*}[\mathbf{x}^\alpha] \right| \leq 1$$

Together with the bound $\mathbb{E}_{(\mathbf{x}, y) \sim D^*}[(\langle \mathbf{v}, \mathbf{x} \rangle)^k] \leq (K_3 k)^{k/2}$, the above implies that $\mathbb{E}_{(\mathbf{x}, y) \sim D}[(\langle \mathbf{v}, \mathbf{x} \rangle)^k] \leq (C_1 k)^{k/2}$ for some constant $C_1$.

Now, we need to show that if the elements of $S$ are chosen i.i.d. from $D^*$, and $|S| \geq \left( d^k, \left( \log \frac{1}{\delta} \right)^k \right)^{C_1}$ then the tester above accepts with probability at least $1 - \delta$. Consider any specific multi-index $\alpha \in \mathbb{Z}^d_{\geq 0}$

with $|\alpha| = k$. Now, by Proposition A.2(f) we have the following:

$$\mathbb{E}_{\mathbf{x} \sim D^*}\left[\left(\mathbf{x}^\alpha - \mathbb{E}_{\mathbf{z} \sim D^*}[\mathbf{z}^\alpha]\right)^{2\log(1/\delta)}\right] \leq \sum_{\ell=0}^{2\log(1/\delta)} \left(\mathbb{E}_{\mathbf{x} \sim D^*}(\mathbf{x}^\alpha)^\ell\right) \cdot \left(\mathbb{E}_{\mathbf{z} \sim D^*}[\mathbf{z}^\alpha]\right)^{2\log(1/\delta)-\ell}$$

$$\leq \sum_{\ell=0}^{2\log(1/\delta)} (K_4 \ell k)^{\ell k/2}(K_4 k)^{k(2\log(1/\delta)-\ell)/2}$$

$$\leq 2\log(1/\delta)(2K_4\log(1/\delta)k)^{\log(1/\delta)k}$$

This, together with Markov's inequality implies that

$$\mathbb{P}\left[\left|\frac{1}{|S|}\sum_{\mathbf{x} \in S}\mathbf{x}^\alpha - \mathbb{E}_{\mathbf{x} \sim D^*}[\mathbf{x}^\alpha]\right| > \frac{1}{d^k}\right] \leq \left(\frac{d^k(3K_4 k \log(1/\delta))^{k/2}}{|S|}\right)^{2\log(1/\delta)}$$

Since $S$ is obtained by taking at least $|S| \geq \left(d^k, \left(\log\frac{1}{\delta}\right)^k\right)^{C_1}$, for sufficiently large $C_1$ we see that the above is upper-bounded by $\frac{1}{d^k}\delta$. Taking a union bound over all $\alpha \in \mathbb{Z}_{\geq 0}^d$ with $|\alpha| = k$, we see that with probability at least $1 - \delta$ the tester $T_1$ accepts, finishing the proof.

## B.3    Proof of Proposition 3.3

Let $K_1$ be as in part (d) of Proposition A.2. The tester $T_2$ computes the fraction of elements in $S$ that are in $T$. If this fraction is $K_1\sigma/2$-close to $\mathbb{P}_{\mathbf{x} \sim D^*}[|\langle \mathbf{w}, \mathbf{x} \rangle| \leq \sigma]$, the algorithm accepts. The algorithm rejects otherwise.

Now, from (d) of Proposition A.2 we have that $\mathbb{P}_{\mathbf{x} \sim D^*}[|\langle \mathbf{w}, \mathbf{x} \rangle| \leq \sigma] \in [K_1\sigma, K_2\sigma]$. Therefore, if the fraction of elements in $S$ that belong in $T$ is $K_1\sigma/100$-close to $\mathbb{P}_{\mathbf{x} \sim D^*}[|\langle \mathbf{w}, \mathbf{x} \rangle| \leq \sigma]$, then this quantity is in $[K_1\sigma/2, (K_2 + K_1/2)\sigma]$ as required.

Finally, if $|S| \geq \frac{100}{K_1\sigma^2}\log\left(\frac{1}{\delta}\right)$ by standard Hoeffding bound, with probability at least $1 - \delta$ we indeed have that the fraction of elements in $S$ that are in $T$ is $K_1\sigma/2$-close to $\mathbb{P}_{\mathbf{x} \sim D^*}[|\langle \mathbf{w}, \mathbf{x} \rangle| \leq \sigma]$.

## B.4    Proof of Proposition 3.4

The tester $T_3$ does the following:

1. Runs the tester $T_2$ from Proposition 3.3. If $T_2$ rejects, $T_3$ rejects as well.

2. Let $S_{|T}$ be the set of elements in $S$ for which $\mathbf{x} \in T$.

3. Let $k = \tilde{O}(1/\tau^2)$ be chosen as in Proposition 3.1.

4. For all $\alpha \in \mathbb{Z}_{\geq 0}^d$ with $|\alpha| = k$:

   (a) Compute the corresponding moment $\mathbb{E}_{(\mathbf{x},y) \sim D}[\mathbf{x}^\alpha \mid \mathbf{x} \in T] := \frac{1}{|S_{|T}|}\sum_{\mathbf{x} \in S_{|T}}\mathbf{x}^\alpha$.

   (b) If $\left|\mathbb{E}_{(\mathbf{x},y) \sim D}[\mathbf{x}^\alpha \mid \mathbf{x} \in T] - \mathbb{E}_{\mathbf{x} \sim D^*}[\mathbf{x}^\alpha \mid \mathbf{x} \in T]\right| > \frac{\tau}{d^k} \cdot d^{-\tilde{O}(k)}$ then reject, where the polylogarithmic in $d^{-\tilde{O}(k)}$ is chosen to satisfy the additive slack condition in Proposition 3.1.

5. If all the checks above passed, accept.

First, we argue that if the checks above pass, then Equations 3.3 and 3.4 will hold. If the tester passes, Equation 3.3 follows immediately from the guarantees in step (4b) of $T_3$ together with Proposition 3.1. Equation 3.4, in turn, is proven as follows:

$$\left|\mathbb{E}_{(\mathbf{x},y) \sim D}[(\langle \mathbf{v}, \mathbf{x} \rangle)^2] - \mathbb{E}_{(\mathbf{x},y) \sim D^*}[(\langle \mathbf{v}, \mathbf{x} \rangle)^2]\right| \leq \sum_{\alpha:|\alpha|=2}\left|\mathbb{E}_{(\mathbf{x},y) \sim D}[\mathbf{x}^\alpha] - \mathbb{E}_{\mathbf{x} \sim D^*}[\mathbf{x}^\alpha]\right|$$

$$\leq d^2 \cdot \max_{\alpha:|\alpha|=2}\left|\mathbb{E}_{(\mathbf{x},y) \sim D}[\mathbf{x}^\alpha] - \mathbb{E}_{\mathbf{x} \sim D^*}[\mathbf{x}^\alpha]\right| \leq \tau$$

571 Now, we need to show that if the elements of $S$ are chosen i.i.d. from $D^*$, and $|S| \geq \dots$ then the
572 tester above accepts with probability at least $1 - \delta$. Consider any specific mult-index $\alpha \in \mathbb{Z}_{\geq 0}^d$ with
573 $|\alpha| = k$. Now, by Proposition A.2(f) we have for any positive integer $\ell$ the following:

$$\mathop{\mathbb{E}}_{\mathbf{x} \sim D^*} \left[ \left| (\mathbf{x}^\alpha)^\ell \right| \right] \leq (K_4 \ell k)^{k/2}$$

574 But by Proposition A.2(d) we have that $\mathbb{P}_{\mathbf{x} \sim D^*}[\mathbf{x} \in T] = \mathbb{P}_{\mathbf{x} \sim D^*}[|\langle \mathbf{x}, \mathbf{w} \rangle| \leq \sigma] \geq K_1 \sigma$. Therefore,
575 the density of the distribution $D^*_{|T}$ (which is defined as the distribution one obtains by taking $D^*$ and
576 conditioning on $\mathbf{x} \in T$) is upper bounded by the product of the density of the distribution $D^*$ and
577 $\frac{1}{K_1 \sigma}$. This allows us to bound

$$\mathop{\mathbb{E}}_{\mathbf{x} \sim D^*} \left[ \left| (\mathbf{x}^\alpha)^\ell \right| \mid \mathbf{x} \in T \right] \leq \frac{1}{K_1 \sigma} \mathop{\mathbb{E}}_{\mathbf{x} \sim D^*} \left[ \left| (\mathbf{x}^\alpha)^\ell \right| \right] \leq \frac{(K_4 \ell k)^{k/2}}{K_1 \sigma}$$

578 This implies that

$$\mathop{\mathbb{E}}_{\mathbf{x} \sim D^*} \left[ \left( \mathbf{x}^\alpha - \mathop{\mathbb{E}}_{\mathbf{z} \sim D^*} [\mathbf{z}^\alpha \mid \mathbf{z} \in T] \right)^{2 \log(1/\delta)} \mid \mathbf{x} \in T \right]$$

$$\leq \sum_{\ell=0}^{2 \log(1/\delta)} \left( \mathop{\mathbb{E}}_{\mathbf{x} \sim D^*} \left[ (\mathbf{x}^\alpha)^\ell \mid \mathbf{x} \in T \right] \right) \cdot \left( \mathop{\mathbb{E}}_{\mathbf{x} \sim D^*} [(\mathbf{x}^\alpha \mid \mathbf{x} \in T]) \right)^{2 \log(1/\delta) - \ell}$$

$$\leq \frac{1}{(K_1 \sigma)^{2 \log(1/\delta)}} \sum_{\ell=0}^{2 \log(1/\delta)} (K_4 \ell k)^{\ell k/2} (K_4 k)^{k(2 \log(1/\delta) - \ell)/2}$$

$$\leq \frac{1}{(K_1 \sigma)^{2 \log(1/\delta)}} 2 \log(1/\delta) (2 K_4 \log(1/\delta) k)^{\log(1/\delta) k}$$

579 This, together with Markov's inequality implies that

$$\mathbb{P} \left[ \left| \frac{1}{|S|} \sum_{\mathbf{x} \in S} \mathbf{x}^\alpha - \mathop{\mathbb{E}}_{\mathbf{x} \sim D^*} [\mathbf{x}^\alpha] \right| > \frac{\tau}{d^k} \cdot d^{-\tilde{O}(k)} \right] \leq \left( \frac{d^{\tilde{O}(k)} (3 K_4 k \log(1/\delta))^{k/2}}{K_1 \sigma |S_{|T}| \tau} \right)^{2 \log(1/\delta)}$$

580 Now, recall that the tester $T_2$ in step (1) accepted, we have $|S_{|T}| \geq \frac{1}{C_2 \sigma} |S|$. Since $S$ is obtained by
581 taking at least $|S| \geq \left( \frac{1}{\tau} \cdot \frac{1}{\sigma} \cdot d^{\frac{1}{\tau^2} \log^{C_5} \left( \frac{1}{\tau} \right)} \cdot \left( \log \frac{1}{\delta} \right)^{\frac{1}{\tau^2} \log^{C_5} \left( \frac{1}{\tau} \right)} \right)^{C_5}$, for sufficiently large $C_5$ we see
582 that the expression above is upper-bounded by $\frac{1}{d^k} \delta$. Taking a union bound over all $\alpha \in \mathbb{Z}_{\geq 0}^d$ with
583 $|\alpha| = k$, we see that with probability at least $1 - \delta$ the tester $T_3$ accepts, finishing the proof.

## C  Proofs from Section 4

585 We first present the following Proposition, which ensures that we can form a loss function with certain
586 desired properties.

587 **Proposition C.1.** *There are constants $c, c' > 0$, such that for any $\sigma > 0$, there exists a continuously*
588 *differentiable function $\ell_\sigma : \mathbb{R} \to [0, 1]$ with the following properties.*

589   *1. For any $t \in [-\sigma/6, \sigma/6]$, $\ell_\sigma(t) = \frac{1}{2} + \frac{t}{\sigma}$.*

590   *2. For any $t > \sigma/2$, $\ell_\sigma(t) = 1$ and for any $t < -\sigma/2$, $\ell_\sigma(t) = 0$.*

591   *3. For any $t \in \mathbb{R}$, $\ell'_\sigma(t) \in [0, c/\sigma]$, $\ell'_\sigma(t) = \ell'_\sigma(-t)$ and $|\ell''_\sigma(t)| \leq c'/\sigma^2$.*

592 *Proof.* We define $\ell_\sigma$ as follows.

$$\ell_\sigma(t) = \begin{cases} \frac{t}{\sigma} + \frac{1}{2}, & \text{if } |t| \leq \frac{\sigma}{6} \\ 1, & \text{if } t > \frac{\sigma}{2} \\ 0, & \text{if } t < \frac{-\sigma}{2} \\ \ell^+(t), t \in \left( \frac{\sigma}{6}, \frac{\sigma}{2} \right] \\ \ell^-(t), t \in \left[ -\frac{\sigma}{2}, -\frac{\sigma}{6} \right) \end{cases}$$

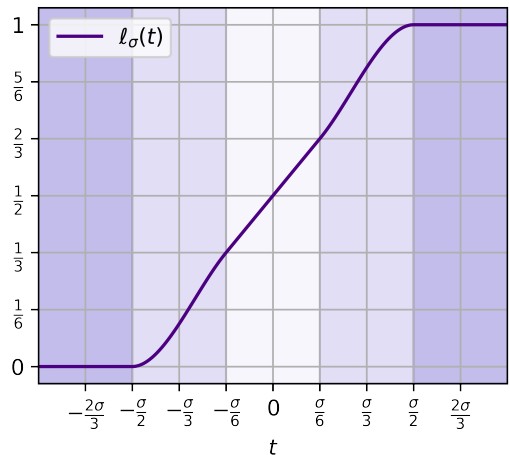

Figure 2: The function $\ell_\sigma$ used to smoothly approximate the ramp.

for some appropriate functions $\ell^+, \ell^-$. It is sufficient that we pick $\ell^+$ satisfying the following conditions (then $\ell^-$ would be defined symmetrically, i.e., $\ell^-(t) = 1 - \ell^+(-t)$).

- $\ell^+(\sigma/2) = 1$ and $\ell^{+\prime}(\sigma/2) = 0$.

- $\ell^+(\sigma/6) = 2/3$ and $\ell^{+\prime}(\sigma/6) = 1/\sigma$.

- $\ell^{+\prime\prime}$ is defined and bounded, except, possibly on $\sigma/6$ and/or $\sigma/2$.

We therefore need to satisfy four equations for $\ell^+$. So we set $\ell^+$ to be a degree 3 polynomial: $\ell^+(t) = a_1 t^3 + a_2 t^2 + a_3 t + a_4$. Whenever $\sigma > 0$, the system has a unique solution that satisfies the desired inequalities. In particular, we may solve the equation to get $a_1 = -9/\sigma^3, a_2 = 15/(2\sigma^2), a_3 = -3/(4\sigma)$ and $a_4 = 5/8$. For the resulting function (see Figure 2 below and Figure 4 in the appendix) we have that there are constants $c, c' > 0$ such that $\ell^{+\prime}(t) \in [0, c/\sigma]$ and $|\ell^{+\prime\prime}(t)| \leq c'/\sigma^2$ for any $t \in [\sigma/6, \sigma/2]$. $\qquad\square$

## C.1 Proof of Lemma 4.3

We will prove the contrapositive of the claim, namely, that there are constants $c_1, c_2, c_3 > 0$ such that if $\angle(\mathbf{w}, \mathbf{w}^*), \angle(-\mathbf{w}, \mathbf{w}^*) > \frac{c_3}{\sqrt{1-2\eta}} \cdot \sigma$, and $\tau \leq c_2$, then $\|\nabla_\mathbf{w} \mathcal{L}_\sigma(\mathbf{w})\|_2 \geq c_1(1 - 2\eta)$.

Consider the case where $\angle(\mathbf{w}, \mathbf{w}^*) < \pi/2$ (otherwise, perform the same argument for $-\mathbf{w}$). Let $\mathbf{v}$ be a unit vector orthogonal to $\mathbf{w}$ that can be expressed as a linear combination of $\mathbf{w}$ and $\mathbf{w}^*$ and for which $\langle \mathbf{v}, \mathbf{w}^* \rangle = 0$. Then $\{\mathbf{v}, \mathbf{w}\}$ is an orthonormal basis for $V = \text{span}(\mathbf{w}, \mathbf{w}^*)$. For any vector $\mathbf{x} \in \mathbb{R}^d$, we will use the following notation: $x_\mathbf{w} = \langle \mathbf{w}, \mathbf{x} \rangle$, $x_\mathbf{v} = \langle \mathbf{v}, \mathbf{x} \rangle$. It follows that $\text{proj}_V(\mathbf{x}) = x_\mathbf{w}\mathbf{w} + x_\mathbf{v}\mathbf{v}$, where $\text{proj}_V$ is the operator that orthogonally projects vectors on $V$.

Using the fact that $\nabla_\mathbf{w}(\langle \mathbf{w}, \mathbf{x} \rangle / \|\mathbf{w}\|_2) = \mathbf{x} - \langle \mathbf{w}, \mathbf{x} \rangle \mathbf{w} = \mathbf{x} - x_\mathbf{w}\mathbf{w}$ for any $\mathbf{w} \in \mathbb{S}^{d-1}$, the interchangeability of the gradient and expectation operators and the fact that $\ell'_\sigma$ is an even function we get that

$$\nabla_\mathbf{w} \mathcal{L}_\sigma(\mathbf{w}) = \mathbb{E}\left[ -\ell'_\sigma(|\langle \mathbf{w}, \mathbf{x} \rangle|) \cdot y \cdot (\mathbf{x} - x_\mathbf{w}\mathbf{w}) \right]$$

Since the projection operator $\text{proj}_V$ is a contraction, we have $\|\nabla_\mathbf{w} \mathcal{L}_\sigma(\mathbf{w})\|_2 \geq \|\text{proj}_V \nabla_\mathbf{w} \mathcal{L}_\sigma(\mathbf{w})\|_2$, and we can therefore restrict our attention to a simpler, two dimensional problem. In particular, since

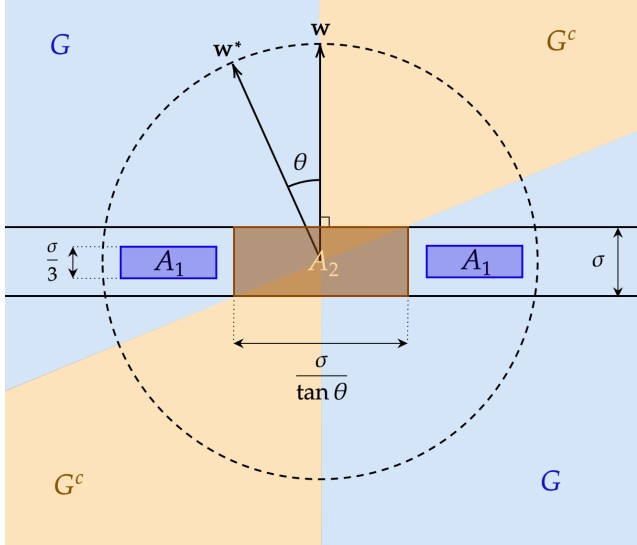

Figure 3: Critical regions in the proofs of main structural lemmas (Lemmas 4.3, 5.2). We analyze the contributions of the regions labeled $A_1, A_2$ to the quantities $A_1, A_2$ in the proofs. Specifically, the regions $A_1$ (which have height $\sigma/3$ so that the value of $\ell'_\sigma(\mathbf{x_w})$ for any $\mathbf{x}$ in these regions is exactly $1/\sigma$, by Proposition C.1) form a subset of the region $\mathcal{G}$, and their probability mass under $D_\mathcal{X}$ is (up to a multiplicative factor) a lower bound on the quantity $A_1$ (see Eq (C.3)). Similarly, the region $A_2$ is a subset of the intersection of $\mathcal{G}^c$ with the band of height $\sigma$, and has probability mass that is (up to a multiplicative factor) an upper bound on the quantity $A_2$ (see Eq (C.4)).

$\mathrm{proj}_V(\mathbf{x}) = \mathbf{x_w w} + \mathbf{x_v v}$, we get

$$\|\mathrm{proj}_V \nabla_\mathbf{w} \mathcal{L}_\sigma(\mathbf{w})\|_2 = \left\| \mathbb{E}\left[ -\ell'_\sigma(|\mathbf{x_w}|) \cdot y \cdot \mathbf{x_v v} \right] \right\|_2$$
$$= \left| \mathbb{E}\left[ -\ell'_\sigma(|\mathbf{x_w}|) \cdot y \cdot \mathbf{x_v} \right] \right|$$
$$= \left| \mathbb{E}\left[ -\ell'_\sigma(|\mathbf{x_w}|) \cdot \mathrm{sign}(\langle \mathbf{w}^*, \mathbf{x}\rangle) \cdot (1 - 2\,\mathbb{1}\{y \neq \mathrm{sign}(\langle \mathbf{w}^*, \mathbf{x}\rangle)\}) \cdot \mathbf{x_v} \right] \right|$$

Let $F(y, \mathbf{x})$ denote $1 - 2\,\mathbb{1}\{y \neq \mathrm{sign}(\langle \mathbf{w}^*, \mathbf{x}\rangle)\}$. We may write $\mathbf{x_v}$ as $|\mathbf{x_v}| \cdot \mathrm{sign}(\mathbf{x_v})$ and let $\mathcal{G} \subseteq \mathbb{R}^2$ such that $\mathrm{sign}(\mathbf{x_v}) \cdot \mathrm{sign}(\langle \mathbf{w}^*, \mathbf{x}\rangle) = -1$ iff $\mathbf{x} \in \mathcal{G}$. Then, $\mathrm{sign}(\mathbf{x_v}) \cdot \mathrm{sign}(\langle \mathbf{w}^*, \mathbf{x}\rangle) = \mathbb{1}\{\mathbf{x} \notin \mathcal{G}\} - \mathbb{1}\{\mathbf{x} \in \mathcal{G}\}$. We get

$$\| \mathrm{proj}_V \nabla_\mathbf{w} \mathcal{L}_\sigma(\mathbf{w})\|_2 =$$
$$= \left| \mathbb{E}\left[ \ell'_\sigma(|\mathbf{x_w}|) \cdot (\mathbb{1}\{\mathbf{x} \in \mathcal{G}\} - \mathbb{1}\{\mathbf{x} \notin \mathcal{G}\}) \cdot F(y, \mathbf{x}) \cdot |\mathbf{x_v}| \cdot \right] \right| \geq$$
$$\geq \mathbb{E}\left[ \ell'_\sigma(|\mathbf{x_w}|) \cdot \mathbb{1}\{\mathbf{x} \in \mathcal{G}\} \cdot F(y, \mathbf{x}) \cdot |\mathbf{x_v}| \right] - \mathbb{E}\left[ \ell'_\sigma(|\mathbf{x_w}|) \cdot \mathbb{1}\{\mathbf{x} \notin \mathcal{G}\} \cdot F(y, \mathbf{x}) \cdot |\mathbf{x_v}| \right]$$

Let $A_1 = \mathbb{E}[\ell'_\sigma(|\mathbf{x_w}|) \cdot \mathbb{1}\{\mathbf{x} \in \mathcal{G}\} \cdot F(y, \mathbf{x}) \cdot |\mathbf{x_v}|]$ and $A_2 = \mathbb{E}[\ell'_\sigma(|\mathbf{x_w}|) \cdot \mathbb{1}\{\mathbf{x} \notin \mathcal{G}\} \cdot F(y, \mathbf{x}) \cdot |\mathbf{x_v}|]$. (See Figure 3.) Note that $\mathbb{E}_{y|\mathbf{x}}[F(y, \mathbf{x})] = 1 - 2\eta(\mathbf{x}) \in [1 - 2\eta, 1]$, where $1 - 2\eta > 0$. Therefore, we have that $A_1 \geq (1 - 2\eta) \cdot \mathbb{E}[\ell'_\sigma(|\mathbf{x_w}|) \cdot \mathbb{1}\{\mathbf{x} \in \mathcal{G}\} \cdot |\mathbf{x_v}|]$ and $A_2 \leq \mathbb{E}[\ell'_\sigma(|\mathbf{x_w}|) \cdot \mathbb{1}\{\mathbf{x} \notin \mathcal{G}\} \cdot |\mathbf{x_v}|]$.

Note that due to Proposition C.1, $\ell'_\sigma(|\mathbf{x_w}|) \leq c/\sigma$ for some constant $c$ and $\ell'_\sigma(|\mathbf{x_w}|) = 0$ whenever $|\mathbf{x_w}| > \sigma/2$. Therefore, if $\mathcal{U}_2$ is the band $B_\mathbf{w}(\sigma/2) = \{\mathbf{x} : |\mathbf{x_w}| \leq \sigma/2\}$ we have

$$A_2 \leq \frac{c}{\sigma} \cdot \mathbb{E}[\mathbb{1}\{\mathbf{x} \notin \mathcal{G}\} \cdot \mathbb{1}\{\mathbf{x} \in \mathcal{U}_2\} \cdot |\mathbf{x_v}|] \tag{C.1}$$

Moreover, for each individual $\mathbf{x}$, we have $\ell'_\sigma(|\mathbf{x_w}|) \cdot \mathbb{1}\{\mathbf{x} \in \mathcal{G}\} \cdot |\mathbf{x_v}| \geq 0$, due to the properties of $\ell'_\sigma$ (Proposition C.1). Hence, for any set $\mathcal{U}_1 \subseteq \mathbb{R}^d$ we have that

$$A_1 \geq (1 - 2\eta) \cdot \mathbb{E}[\ell'_\sigma(|\mathbf{x_w}|) \cdot \mathbb{1}\{\mathbf{x} \in \mathcal{G}\} \cdot \mathbb{1}\{\mathbf{x} \in \mathcal{U}_1\} \cdot |\mathbf{x_v}|]$$

Setting $\mathcal{U}_1 = B_{\mathbf{w}}(\sigma/6) = \{\mathbf{x} : |\mathbf{x}_{\mathbf{w}}| \le \sigma/6\}$, by Proposition C.1, we get $\ell'_\sigma(|\mathbf{x}_{\mathbf{w}}|) \cdot \mathbb{1}\{\mathbf{x} \in \mathcal{U}_1\} = \frac{1}{\sigma} \cdot \mathbb{1}\{\mathbf{x} \in \mathcal{U}_1\}$.

$$A_1 \ge \frac{1-2\eta}{\sigma} \cdot \mathbb{E}[\mathbb{1}\{\mathbf{x} \in \mathcal{G}\} \cdot \mathbb{1}\{\mathbf{x} \in \mathcal{U}_1\} \cdot |\mathbf{x}_{\mathbf{v}}|] \tag{C.2}$$

We now observe that by the definitions of $\mathcal{G}, \mathcal{U}_1, \mathcal{U}_2$, for any constant $R > 0$, there exist some constants $c', c'' > 0$ such that if $\sigma/\tan\theta < c'R$ (the points in $\mathbb{R}^2$ where $\partial\overline{\mathcal{G}}$ intersects either $\partial\mathcal{U}_1$ or $\partial\mathcal{U}_2$ have projections on $\mathbf{v}$ that are $\Theta(\sigma/\tan\theta)$) we have that

$$\mathbb{1}\{\mathbf{x} \in \mathcal{G}\} \cdot \mathbb{1}\{\mathbf{x} \in \mathcal{U}_1\} \ge \mathbb{1}\{|\mathbf{x}_{\mathbf{v}}| \in [c'R, 2c'R]\} \cdot \mathbb{1}\{\mathbf{x} \in \mathcal{U}_1\} \quad \text{and}$$
$$\mathbb{1}\{\mathbf{x} \in \mathcal{G}\} \cdot \mathbb{1}\{\mathbf{x} \in \mathcal{U}_2\} \le \mathbb{1}\{|\mathbf{x}_{\mathbf{v}}| \le c''\sigma/\tan\theta\} \cdot \mathbb{1}\{\mathbf{x} \in \mathcal{U}_2\}$$

By equations (C.1) and (C.2), we get the following bounds whose graphical representations can be found in Figure 3.

$$A_1 \ge \frac{c'R(1-2\eta)}{\sigma} \cdot \mathbb{E}[\mathbb{1}\{|\mathbf{x}_{\mathbf{v}}| \in [c'R, 2c'R]\} \cdot \mathbb{1}\{\mathbf{x} \in \mathcal{U}_1\}] \tag{C.3}$$

$$A_2 \le \frac{c \cdot c''}{\tan\theta} \cdot \mathbb{E}[\mathbb{1}\{|\mathbf{x}_{\mathbf{v}}| \le c''\sigma/\tan\theta\} \cdot \mathbb{1}\{\mathbf{x} \in \mathcal{U}_2\}] \tag{C.4}$$

So far, we have used no distributional assumptions. Now, consider the corresponding expectations under the target marginal $D^*$ (which we assumed to be strongly log-concave).

$$I_1 = \mathbb{E}_{D^*}[\mathbb{1}\{|\mathbf{x}_{\mathbf{v}}| \in [c'R, 2c'R]\} \cdot \mathbb{1}\{\mathbf{x} \in \mathcal{U}_1\}]$$
$$I_2 = \mathbb{E}_{D^*}[\mathbb{1}\{|\mathbf{x}_{\mathbf{v}}| \le c''\sigma/\tan\theta\} \cdot \mathbb{1}\{\mathbf{x} \in \mathcal{U}_2\}]$$

Any strongly log-concave distribution enjoys the "well-behaved" properties defined by [DKTZ20a], and therefore, if $R$ is picked to be small enough, then $I_1$ and $I_2$ are of order $\Theta(\sigma)$ (due to upper and lower bounds on the two dimensional marginal density over $V$ within constant radius balls – aka anti-anticoncentration and anticoncentration). Moreover, by Proposition A.2, we have $\mathbb{P}[\mathbf{x} \in \mathcal{U}_1]$ and $\mathbb{P}[\mathbf{x} \in \mathcal{U}_2]$ are both of order $\Theta(\sigma)$. Hence we have that there exist constants $c'_1, c'_2 > 0$ such that for the conditional expectations we have

$$\mathbb{E}_{D^*}\big[\mathbb{1}\{|\mathbf{x}_{\mathbf{v}}| \in [c'R, 2c'R]\} \mid \mathbb{1}\{\mathbf{x} \in \mathcal{U}_1\}\big] \ge c'_1$$
$$\mathbb{E}_{D^*}\big[\mathbb{1}\{|\mathbf{x}_{\mathbf{v}}| \le c''\sigma/\tan\theta\} \mid \mathbb{1}\{\mathbf{x} \in \mathcal{U}_2\}\big] \le c'_2$$

By assumption, Property (3.3) holds and, therefore, if $\tau \le c'_1/2, c'_2/2 =: c_2$, we get that

$$\mathbb{E}_{D_{\mathcal{X}}}\big[\mathbb{1}\{|\mathbf{x}_{\mathbf{v}}| \in [c'R, 2c'R]\} \mid \mathbb{1}\{\mathbf{x} \in \mathcal{U}_1\}\big] \ge c'_1/2$$
$$\mathbb{E}_{D_{\mathcal{X}}}\big[\mathbb{1}\{|\mathbf{x}_{\mathbf{v}}| \le c''\sigma/\tan\theta\} \mid \mathbb{1}\{\mathbf{x} \in \mathcal{U}_2\}\big] \le c'_2/2$$

Moreover, by Property (3.2), we have that (under the true marginal) $\mathbb{P}[\mathbf{x} \in \mathcal{U}_1]$ and $\mathbb{P}[\mathbf{x} \in \mathcal{U}_2]$ are both $\Theta(\sigma)$. Hence, in total, we get that for some constants $\tilde{c}_1, \tilde{c}_2$, we have

$$A_1 \ge \tilde{c}_1 \cdot (1 - 2\eta)$$
$$A_2 \le \tilde{c}_2 \cdot \frac{\sigma}{\tan\theta}$$

Hence, if we pick $\sigma = \Theta((1-2\eta)\tan\theta)$, we get the desired result.

## C.2   Proof of Proposition 4.4

For the following all the probabilities and expectations are over $D_{\mathcal{X}\mathcal{Y}}$. First we observe that

$$\mathbb{P}[y \ne \mathrm{sign}(\langle\mathbf{w}, \mathbf{x}\rangle)] \le \mathbb{P}[y \ne \mathrm{sign}(\langle\mathbf{w}, \mathbf{x}\rangle) \cap y = \mathrm{sign}(\langle\mathbf{w}^*, \mathbf{x}\rangle)] + \mathbb{P}[y \ne \mathrm{sign}(\langle\mathbf{w}^*, \mathbf{x}\rangle)] \le$$
$$\le \mathbb{P}[\mathrm{sign}(\langle\mathbf{w}, \mathbf{x}\rangle) \ne \mathrm{sign}(\langle\mathbf{w}^*, \mathbf{x}\rangle)] + \mathsf{opt}.$$

Then, we observe that by assumption that $D_{\mathcal{X}\mathcal{Y}}$ satisfies Property (3.2), we have

$$\mathbb{P}[|\langle\mathbf{w}, \mathbf{x}\rangle| \le \sigma] \le C_3\sigma$$

and that

$$\mathbb{P}[\text{sign}(\langle \mathbf{w}, \mathbf{x} \rangle) \neq \text{sign}(\langle \mathbf{w}^*, \mathbf{x} \rangle) \cap |\langle \mathbf{w}, \mathbf{x} \rangle| > \sigma] \leq \mathbb{P}\left[|\langle \mathbf{v}, \mathbf{x} \rangle| \geq \frac{\sigma}{\tan \theta}\right],$$

where $\mathbf{v}$ is some vector perpendicular to $\mathbf{w}$. Using Markov's inequality, we get

$$\mathbb{P}\left[|\langle \mathbf{v}, \mathbf{x} \rangle| \geq \frac{\sigma}{\tan \theta}\right] \leq \frac{(\tan \theta)^k}{\sigma^k} \cdot \mathbb{E}[|\langle \mathbf{v}, \mathbf{x} \rangle|^k].$$

But, by assumption that $D_{\mathcal{X}\mathcal{Y}}$ satisfies Property (3.1), there is some constant $C_1 > 0$ such that $\mathbb{E}[|\langle \mathbf{v}, \mathbf{x} \rangle|^k] \leq (C_1 k)^{k/2}$. Thus

$$\begin{aligned}
\mathbb{P}[\text{sign}(\langle \mathbf{w}, \mathbf{x} \rangle) \neq \text{sign}(\langle \mathbf{w}^*, \mathbf{x} \rangle)] &\leq \mathbb{P}[|\langle \mathbf{w}, \mathbf{x} \rangle| \leq \sigma] \\
&\quad + \mathbb{P}[\text{sign}(\langle \mathbf{w}, \mathbf{x} \rangle) \neq \text{sign}(\langle \mathbf{w}^*, \mathbf{x} \rangle) \cap |\langle \mathbf{w}, \mathbf{x} \rangle| > \sigma] \\
&\leq C_3 \sigma + \frac{(C_1 k)^{k/2} (\tan \theta)^k}{\sigma^k}.
\end{aligned}$$

By picking $\sigma$ appropriately in order to balance the two terms (note that this is a different $\sigma$ than the one in Lemma 4.3), we get the desired result.

# D    Proofs from Section 5

## D.1    Proof of Theorem 5.1

We will follow the same steps as for proving Theorem 4.1. Once more, we draw a sufficiently large sample so that our testers are ensured to accept with high probability when the true marginal is indeed the target marginal $D^*$ and so that we have generalization, i.e. the guarantee that any approximate minimizer of the empirical error (error on the uniform empirical distribution over the sample drawn) is also an approximate minimizer of the true error. The algorithm we use is once more Algorithm 1, but this time we make multiple calls for different parameters $\sigma$ (and we run $T_1$ with higher $k$, as we will see shortly) and reject if any of these calls rejects. If we accept, we output the output of the execution of Algorithm 1 with the minimum empirical error.

The main difference between the Massart noise case and the agnostic case is that in the former we were able to pick $\sigma$ arbitrarily small, while in the latter we face a more delicate tradeoff. To balance competing contributions to the gradient norm, we must ensure that $\sigma$ is at least $\Theta(\text{opt})$ while also ensuring that it is not too large. And since we do not know the value of opt, we will need to search over a space of possible values for $\sigma$ that is only polynomially large in relevant parameters (similar to the approach of [DKTZ20b]). In our case, we may sparsify the space $(0, 1]$ of possible values for $\sigma$ up to accuracy $\Theta((\frac{\epsilon}{\sqrt{k}})^{1+1/k})$ and form a list of $\text{poly}(k/\epsilon)$ possible values for $\sigma$, one of which will satisfy $c_1 \sigma - \Theta((\frac{\epsilon}{\sqrt{k}})^{1+1/k}) \leq \text{opt} \leq c_1 \sigma$. hence, we perform the same (testing-learning) process for each of the possible values of $\sigma$ and get a list of candidate vectors which is still of polynomial size.

The final step is, again, to use Proposition 4.4, after running tester $T_1$ with parameter $k$ (Proposition 3.2) and tester $T_2$ with appropriate parameters for each of the candidate weight vectors. We get that our list contains a vector $\mathbf{w}$ with

$$\mathbb{P}_{D_{\mathcal{X}\mathcal{Y}}}[y \neq \text{sign}(\langle \mathbf{w}, \mathbf{x} \rangle)] \leq \text{opt} + c \cdot k^{1/2} \cdot \theta^{1-1/(k+1)},$$

where $\measuredangle(\mathbf{w}, \mathbf{w}^*) \leq \theta := c_2 \sigma$ for $\sigma$ such that $c_1 \sigma - \Theta((\frac{\epsilon}{\sqrt{k}})^{1+1/k}) \leq \text{opt} \leq c_1 \sigma$.

$$\mathbb{P}_{D_{\mathcal{X}\mathcal{Y}}}[y \neq \text{sign}(\langle \mathbf{w}, \mathbf{x} \rangle)] \leq \text{opt} + c\sqrt{k} \cdot \left(\frac{c_2}{c_1}\text{opt} + \Theta\left(\left(\frac{\epsilon}{\sqrt{k}}\right)^{1+\frac{1}{k}}\right)\right)^{1-\frac{1}{k+1}} \leq O(\sqrt{k} \cdot \text{opt}^{1-\frac{1}{k+1}}) + \epsilon.$$

However, we do not know which of the weight vectors in our list is the one guaranteed to achieve small error. In order to discover this vector, we estimate the probability of error of each of the corresponding halfspaces (which can be done efficiently, due to Hoeffding's bound) and pick the one with the smallest error. This final step does not require any distributional assumptions and we do not need to perform any further tests.

In order to obtain our $\tilde{O}(\mathsf{opt})$ quasipolynomial time guarantee, observe first that we may assume without loss of generality that $\mathsf{opt} \geq 1/d^C$ for some $C$; if instead $\mathsf{opt} = o(1/d^2)$, say, then a sample of $O(d)$ points will with high probability be noiseless, and so simple linear programming will recover a consistent halfspace that will generalize. Moreover, we may assume that $\mathsf{opt} \leq 1/10$, since otherwise achieving $O(\mathsf{opt})$ is trivial (we may output an arbitrary halfspace). Let us adapt our algorithm so that we run tester $T_1$ (see Proposition 3.2) multiple times for all $k = 1, 2, \ldots, \lceil \log^2 d \rceil$ (this only changes our time and sample complexity by a $\mathrm{polylog}(d)$ factor). Then Proposition 4.4 holds for some $k^*$ such that $k^* \in [\log(1/\mathsf{opt}), 2\log(1/\mathsf{opt})]$, since the interval has length at least 1 (and therefore it contains some integer) and $2\log(1/\mathsf{opt}) \leq 2C \log d \leq \log^2 d$ (for large enough $d$). Therefore, by picking the best candidate we get a guarantee of order

$$
\begin{aligned}
\sqrt{k^*} \cdot \mathsf{opt}^{1-1/k^*} &= \sqrt{k^*} \cdot \mathsf{opt}^{-1/k^*} \mathsf{opt} \\
&= \sqrt{k^*} \cdot 2^{\frac{1}{k^*} \log \frac{1}{\mathsf{opt}}} \cdot \mathsf{opt} \\
&\leq \sqrt{2\log(1/\mathsf{opt})} \cdot 2 \cdot \mathsf{opt} \qquad (\text{since } \log(1/\mathsf{opt}) \leq k^* \leq 2\log(1/\mathsf{opt})) \\
&= \tilde{O}(\mathsf{opt}) \, .
\end{aligned}
$$

This concludes the proof of Theorem 5.1.

## D.2 Proof of Lemma 5.2

In the agnostic case, the proof is analogous to the proof of Lemma 4.3. However, in this case, the difference is that the random variable $F(y, \mathbf{x}) = 1 - 2\, \mathbb{1}\{y \neq \mathrm{sign}(\langle \mathbf{w}^*, \mathbf{x} \rangle)\}$ does not have conditional expectation on $\mathbf{x}$ that is lower bounded by a constant. Instead, we need to consider an additional term $A_3$ correcponding to the part $2\, \mathbb{1}\{y \neq \mathrm{sign}(\langle \mathbf{w}^*, \mathbf{x} \rangle)\}$ and the term $A_1$ will not be scaled by the factor $(1 - 2\eta)$ as in Lemma 4.3. Hence, with similar arguments we have that

$$
\|\nabla_{\mathbf{w}} \mathcal{L}_\sigma(\mathbf{w})\|_2 \geq A_1 - A_2 - A_3 \, ,
$$

where $A_1 \geq \tilde{c}_1$, $A_2 \leq \tilde{c}_2 \cdot \frac{\sigma}{\tan \theta}$ and (using properties of $\ell'_\sigma$ as in Lemma 4.3 and the Cauchy-Schwarz inequality)

$$
\begin{aligned}
A_3 = 2\, \mathbb{E}[\ell'_\sigma(|\mathbf{x}_{\mathbf{w}}|) \cdot \mathbb{1}\{\mathbf{x} \in \mathcal{G}\} \cdot \mathbb{1}\{y \neq \mathrm{sign}(\langle \mathbf{w}, \mathbf{x} \rangle)\} \cdot |\mathbf{x}_{\mathbf{v}}|] &\leq \\
\leq \frac{2c}{\sigma} \cdot \mathbb{E}[\mathbb{1}\{\mathbf{x} \in \mathcal{U}_2\} \cdot \mathbb{1}\{y \neq \mathrm{sign}(\langle \mathbf{w}, \mathbf{x} \rangle)\} \cdot |\mathbf{x}_{\mathbf{v}}|] &\leq \\
\leq \frac{2c}{\sigma} \cdot \sqrt{\mathbb{E}[\mathbb{1}\{\mathbf{x} \in \mathcal{U}_2\} \cdot (\mathbf{x}_{\mathbf{v}})^2]} \cdot \sqrt{\mathbb{E}[\mathbb{1}\{y \neq \mathrm{sign}(\langle \mathbf{w}, \mathbf{x} \rangle)\}]} &= \\
= \frac{2c\sqrt{\mathsf{opt}}}{\sigma} \cdot \sqrt{\mathbb{E}[\langle \mathbf{v}, \mathbf{x} \rangle^2 \mid \mathbf{x} \in \mathcal{U}_2] \cdot \mathbb{P}[\mathbf{x} \in \mathcal{U}_2]} \, . &
\end{aligned}
$$

Similarly to our approach in the proof of Lemma 4.3, we can use the assumed properties (3.2) and (3.4) to get that

$$
A_3 \leq \tilde{c}_3 \frac{\sqrt{\mathsf{opt}}}{\sqrt{\sigma}} \, ,
$$

which gives that in order for the gradient loss to be small, we require $\mathsf{opt} \leq \Theta(\sigma)$.

## D.3 Proof of Theorem 5.3

Before presenting the proof of Theorem 5.3, we prove the following Proposition, which is, essentially, a stronger version of Proposition 4.4 for the specific case when the target marginal distribution $D^*$ is the standard multivariate Gaussian distribution. Proposition D.1 is important to get an $O(\mathsf{opt})$ guarantee for the case where the target distribution is the standard Gaussian.

**Proposition D.1.** *Let $D_{\mathcal{X}\mathcal{Y}}$ be a distribution over $\mathbb{R}^d \times \{\pm 1\}$, $\mathbf{w}^* \in \arg\min_{\mathbf{w} \in \mathbb{S}^{d-1}} \mathbb{P}_{D_{\mathcal{X}\mathcal{Y}}}[y \neq \mathrm{sign}(\langle \mathbf{w}, \mathbf{x} \rangle)]$ and $\mathbf{w} \in \mathbb{S}^{d-1}$. Let $\theta \geq \angle(\mathbf{w}, \mathbf{w}^*)$ and suppose that $\theta \in [0, \pi/4]$. Then, for a sufficiently large constant $C$, there is a tester that given $\delta \in (0, 1)$, $\theta$, $\mathbf{w}$ and a set $S$ of samples from $D_{\mathcal{X}}$ with size at least $\left(\frac{d}{\theta} \log \frac{1}{\delta}\right)^C$, runs in time $\mathrm{poly}\left(\frac{1}{\theta}, d, \log \frac{1}{\delta}\right)$ and with probability $1 - \delta$ satisfies the following specifications:*

- *If the distribution $D_{\mathcal{X}}$ is $\mathcal{N}(0, I_d)$, the tester accepts.*

718 • *If the tester accepts, then we have:*

$$\Pr_{\mathbf{x}\sim S}[\mathrm{sign}(\langle \mathbf{w}^*, \mathbf{x}\rangle) \neq \mathrm{sign}(\langle \mathbf{w}, \mathbf{x}\rangle)] \leq O(\theta)$$

719 *Proof.* The testing algorithm does the following:

720   1. **Given:** Integer $d$, set $S \subset \mathbb{R}^d$, $\mathbf{w} \in \mathbb{S}^{d-1}$, $\theta \in (0, \pi/4]$ and $\delta \in (0, 1)$.

721   2. Let $\mathrm{proj}_{\perp \mathbf{w}} : \mathbb{R}^d \to \mathbb{R}^{d-1}$ denote the operator that projects a vector $\mathbf{x} \in \mathbb{R}^d$ to it's projection
722     into the $(d-1)$-dimensional subspace of $\mathbb{R}^d$ that is orthogonal to $\mathbf{w}$.

723   3. For $i$ in $\left\{0, \pm 1, \cdots, \pm \frac{\sqrt{2 \log \frac{1}{\theta}}}{\theta}\right\}$

724     (a) $S_i \leftarrow \{\mathbf{x} \in S : \langle \mathbf{w}, \mathbf{x}\rangle \in [i\theta, (i+1)\theta]\}$
725     (b) If $\frac{|S_i|}{|S|} > 2\theta$, then reject.
726     (c) If $\left\| \frac{1}{|S_i|} \sum_{\mathbf{x} \in S_i} (\mathrm{proj}_{\perp \mathbf{w}}(\mathbf{x}))(\mathrm{proj}_{\perp \mathbf{w}}(\mathbf{x}))^T - I_{(d-1)} \right\|_{\mathrm{op}} > 0.1$, reject.

727   4. If $\frac{1}{|S|} \sum_{\mathbf{x} \in S} \mathbb{1}_{|\langle \mathbf{w}, \mathbf{x}\rangle| > \sqrt{2 \log \frac{1}{\theta}}} > 5\theta$, then reject.

728   5. If reached this step, accept.

729 If the tester accepts, then we have the following properties for some sufficiently large constant $C' > 0$.
730 For the following, consider the vector $\mathbf{v} \in \mathbb{R}^d$ to be the vector that is perpendicular to $\mathbf{w}$, lies within
731 the plane defined by $\mathbf{w}$ and $\mathbf{w}^*$ and $\langle \mathbf{v}, \mathbf{w}^*\rangle \leq 0$.

732   1. $\mathbb{P}_{\mathbf{x}\sim S}[|\langle \mathbf{w}, \mathbf{x}\rangle| \in [\theta i, \theta(i+1)]] \leq C'\theta$, for any $i \in \left\{0, \pm 1, \ldots, \pm \frac{1}{\theta}\sqrt{2 \log \frac{1}{\theta}}\right\}$.

733   2. $\mathbb{P}_{\mathbf{x}\sim S_i}\left[|\langle \mathbf{v}, \mathbf{x}\rangle| > \frac{\theta}{\tan \theta} \cdot i\right] \leq C'/i^2$, for any $i \in \left\{0, \pm 1, \ldots, \pm \frac{1}{\theta}\sqrt{2 \log \frac{1}{\theta}}\right\}$.

734   3. $\mathbb{P}_{\mathbf{x}\sim S}\left[|\langle \mathbf{w}, \mathbf{x}\rangle| \geq \sqrt{2 \log \frac{1}{\theta}}\right] \leq C'\theta$.

735 Then, for $k = \frac{1}{\theta}\sqrt{2 \log \frac{1}{\theta}}$ and $\mathrm{Strip}_i = \{\mathbf{x} \in \mathbb{R}^d : \langle \mathbf{w}, \mathbf{x}\rangle| \in [\theta i, \theta(i+1)]\}$, we have that

$$\Pr_{\mathbf{x}\sim S}[\mathrm{sign}(\langle \mathbf{w}, \mathbf{x}\rangle) \neq \mathrm{sign}(\langle \mathbf{w}^*, \mathbf{x}\rangle)] \leq$$

$$\sum_{i=-k}^{k} \mathbb{P}_{\mathbf{x}\sim S}[\mathbf{x} \in \mathrm{Strip}_i] \cdot \mathbb{P}_{\mathbf{x}\sim S}\left[|\langle \mathbf{v}, \mathbf{x}\rangle| > \frac{\theta}{\tan \theta} \cdot i \,\Big|\, \mathbf{x} \in \mathrm{Strip}_i\right] + \mathbb{P}_{\mathbf{x}\sim S}\left[|\langle \mathbf{w}, \mathbf{x}\rangle| \geq \sqrt{2 \log \frac{1}{\theta}}\right] \leq$$

$$\sum_{i=-k}^{k} \frac{|S_i|}{|S|} \cdot \mathbb{P}_{\mathbf{x}\sim S_i}\left[|\langle \mathbf{v}, \mathbf{x}\rangle| > \frac{\theta}{\tan \theta} \cdot i\right] + C'\theta \leq (C')^2 \theta \cdot \left(1 + \sum_{i \neq 0} \frac{2}{i^2}\right) + C'\theta = O(\theta)$$

736 Now, suppose the distribution $D_{\mathcal{X}}$ is indeed the standard Gaussian $\mathcal{N}(0, I_d)$. We would like to show
737 that our tester accepts with probability at least $1 - \delta$. Since $D = \mathcal{N}(0, I_d)$, we see that for $\mathbf{x} \sim D$
738 we have that $\mathbf{x} \cdot \mathbf{w}$ is distributed as $\mathcal{N}(0, 1)$. This implies that

739   • For all $i \in \left\{0, \pm 1, \cdots, \pm \frac{\sqrt{2 \log \frac{1}{\theta}}}{\theta}\right\}$ we have

740     – $\Pr_{\mathbf{x}\sim \mathcal{N}(0, I_d)}[\langle \mathbf{w}, \mathbf{x}\rangle \in [i\theta, (i+1)\theta]] \leq \frac{1}{\sqrt{2\pi}}\theta$
741     – $\Pr_{\mathbf{x}\sim \mathcal{N}(0, I_d)}[\langle \mathbf{w}, \mathbf{x}\rangle \in [i\theta, (i+1)\theta]] \geq \theta \cdot \min_{x \in \left[-\sqrt{2 \log \frac{1}{\theta}}-\theta, \sqrt{2 \log \frac{1}{\theta}}+\theta\right]} \frac{1}{\sqrt{2\pi}} e^{-\frac{x^2}{2}} \geq$
742     $\frac{\theta^2}{10}$

743      • $\Pr_{\mathbf{x} \sim \mathcal{N}(0, I_d)} [\langle \mathbf{w}, \mathbf{x} \rangle \in [i\theta, (i+1)\theta]] \leq \frac{1}{\sqrt{2\pi}}\theta$

744      • $\Pr_{\mathbf{x} \sim \mathcal{N}(0, I_d)} \left[ \langle \mathbf{w}, \mathbf{x} \rangle > 2\sqrt{\log \frac{1}{\theta}} \right] = \int_{2\sqrt{\log \frac{1}{\theta}}}^{\infty} \frac{1}{\sqrt{2\pi}} e^{-\frac{x^2}{2}} \, dx \leq \theta \int_0^{\infty} \frac{1}{\sqrt{2\pi}} e^{-\frac{x^2}{2}} \, dx = \frac{\theta}{2}$

745 Therefore, via the standard Hoeffding bound, we see that for sufficiently large absolute constant $C$
746 we have with probability at least $1 - \frac{\delta}{4}$ over the choice of $S$ that

747      • For all $i \in \left\{ 0, \pm 1, \cdots, \pm \frac{\sqrt{2 \log \frac{1}{\theta}}}{\theta} \right\}$ we have

748          – $\Pr_{\mathbf{x} \sim S} [\langle \mathbf{w}, \mathbf{x} \rangle \in [i\theta, (i+1)\theta]] \leq \theta$
749          – $\Pr_{\mathbf{x} \sim S} [\langle \mathbf{w}, \mathbf{x} \rangle \in [i\theta, (i+1)\theta]] \geq \frac{\theta^2}{20}$

750      • $\Pr_{\mathbf{x} \sim S} \left[ \langle \mathbf{w}, \mathbf{x} \rangle > 2\sqrt{\log \frac{1}{\theta}} \right] \leq \theta$

751      • $\Pr_{\mathbf{x} \sim S} \left[ \langle \mathbf{w}, \mathbf{x} \rangle < -2\sqrt{\log \frac{1}{\theta}} \right] \leq \theta$

752 Finally, we would like to show that conditioned on the above, the probability of rejection in step (3b)
753 is small.

754 **Fact D.2.** *Given a set $S \subset \mathbb{R}^{d-1}$ of i.i.d. samples from $\mathcal{N}(0, I_d)$, with probability at least $1 - $*
755 *$\text{poly}\left( \frac{|S|}{d} \right)$ we have*

$$\left\| \frac{1}{|S|} \sum_{\mathbf{x} \in S} \mathbb{1}_{\langle \mathbf{w}, \mathbf{x} \rangle \in [i\theta, (i+1)\theta]} \mathbf{x} \mathbf{x}^T - I_{(d-1)} \right\|_{op} \leq 0.1$$

756 Now, since each sample $\mathbf{x}_i$ is drawn i.i.d. from $\mathcal{N}(0, I_d)$, we have that $\langle \mathbf{w}, \mathbf{x}_i \rangle$ and $\text{proj}_{\perp \mathbf{w}}(\mathbf{x}_i)$ are
757 all independent from each other for all $i$. Since all the events we conditioned on depend on $\{\langle \mathbf{w}, \mathbf{x}_i \rangle\}$
758 we see that $\{\text{proj}_{\perp \mathbf{w}}(\mathbf{x}_i)\}$ are still distributed as i.i.d. samples from $\mathcal{N}(0, I_{(d-1)})$.

759 Recall that one of the events we have already conditioned on is that $\Pr_{\mathbf{x} \sim S} [\langle \mathbf{w}, \mathbf{x} \rangle \in [i\theta, (i+1)\theta]] \geq$
760 $\frac{\theta^2}{20}$ for all $i \in \left\{ 0, \pm 1, \cdots, \pm \frac{\sqrt{2 \log \frac{1}{\theta}}}{\theta} \right\}$. This allows us to lower bound by $\theta^2 / 20$ the ratio $|S_i| / |S|$.
761 And since, as we described, for all these elements $\mathbf{x}_i$ the vectors $\text{proj}_{\perp \mathbf{w}}(\mathbf{x}_i)$ are distributed as i.i.d.
762 samples from $\mathcal{N}(0, I_{(d-1)})$, we can use Fact D.2 to conclude that for sufficiently large absolute con-
763 stant $C$, when $|S| = \left( \frac{d}{\theta} \log \frac{1}{\delta} \right)^C$ we have with probability $1 - \frac{\delta}{4}$ for all $i \in \left\{ 0, \pm 1, \cdots, \pm \frac{\sqrt{2 \log \frac{1}{\theta}}}{\theta} \right\}$
764 that

$$\left\| \frac{1}{|S_i|} \sum_{\mathbf{x} \in S_i} (\text{proj}_{\perp \mathbf{w}}(\mathbf{x}))(\text{proj}_{\perp \mathbf{w}}(\mathbf{x}))^T - I_{(d-1)} \right\|_{op} \leq 0.1$$

765 Overall, this allows us to conclude that with probability at least $1 - \delta$ the tester accepts. □

766 We now present the proof of Theorem 5.3.

767 In the proof of Theorem 5.1, when the target distribution is the standard Gaussian in $d$ dimensions,
768 we may apply Proposition D.1 (and run the corresponding tester), instead of Proposition 4.4, in order
769 to ensure that our list will contain a vector $\mathbf{w}$ with

$$\Pr_{D_{\mathcal{X}\mathcal{Y}}} [y \neq \text{sign}(\langle \mathbf{w}, \mathbf{x} \rangle)] \leq \Pr_{D_{\mathcal{X}\mathcal{Y}}} [y \neq \text{sign}(\langle \mathbf{w}^*, \mathbf{x} \rangle)] + \Pr_{D_{\mathcal{X}\mathcal{Y}}} [\text{sign}(\langle \mathbf{w}^*, \mathbf{x} \rangle) \neq \text{sign}(\langle \mathbf{w}, \mathbf{x} \rangle)]$$
$$\leq \text{opt} + O(\theta)$$

770 where $\angle(\mathbf{w}, \mathbf{w}^*) \leq \theta := c_2 \sigma$ and $\sigma$ is such that $c_1 \sigma - \Theta(\epsilon) \leq \text{opt} \leq c_1 \sigma$, which gives the desired
771 $O(\text{opt}) + \epsilon$ bound. To get the value of $\sigma$ with the desired property, we once again sparsified the space
772 $(0, 1]$ of possible values for $\sigma$, this time up to accuracy $\Theta(\epsilon)$.

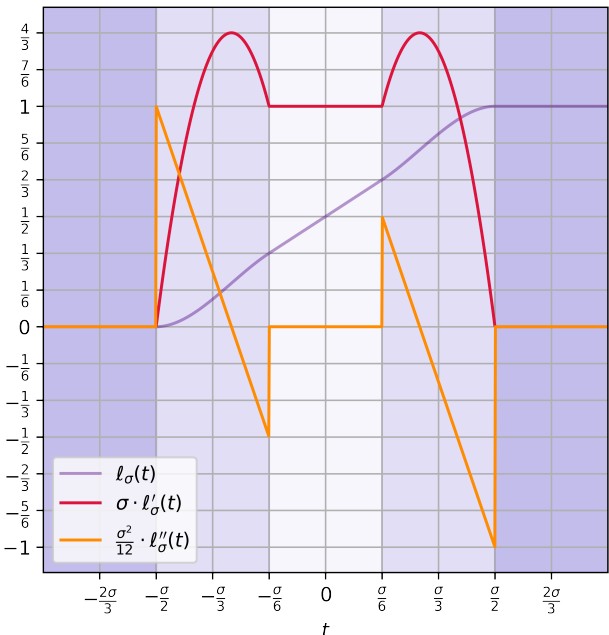

Figure 4: Figure illustrating the (normalized) first two derivatives of the function $\ell_\sigma$ used to define the non convex surrogate loss $\mathcal{L}_\sigma$. The normalization is appropriate since $\ell'_\sigma$ and $\ell''_\sigma$ are homogeneous in $1/\sigma$ and $1/\sigma^2$ respectively. In particular, we see that $\ell'_\sigma \leq \Theta(1/\sigma)$ and $|\ell''_\sigma| \leq \Theta(1/\sigma^2)$ everywhere.