# OpenReview forum: "An Efficient Tester-Learner for Halfspaces"
_NeurIPS.cc/2023/Conference — Submitted to NeurIPS 2023_

### Official Review · Reviewer_Vb7c · 2023-07-05

**Soundness:** 3 good
**Presentation:** 3 good
**Contribution:** 4 excellent
**Rating:** 7
**Confidence:** 4

**Summary:**

The paper introduces an efficient algorithm for learning halfspaces in the testable learning model in which the tester-learner first applies a test on the training data and if the test succeeds the algorithm produces a hypothesis which is guaranteed to be near-optimal. It is required that if the data comes from a target distribution, then the test must succeed with high probability.

The paper considers learning halfspaces in the case where the target distribution is Gaussian (or strongly log-concave) and where the labels are subjected to Massart noise or adversarial noise (i.e., agnostic setting). The paper builds on several ideas from previous papers by Diakonikolas et al.

**Strengths:**

Learning halfspaces is a fundamental problem in machine learning. Even though it is one of the simplest tasks, the problem becomes non-trivial in the presence of label noise. Recently, there has been a lot of interest in developing algorithms for learning halfspaces in several settings (e.g., Massart noise, ...). The submitted paper is the first work that presents an efficient algorithm for learning halfspaces in the testable learning model of Rubinfeld and Vasilyan.

I find the paper to be clear and generally well-written, and I find the results novel and interesting.

**Weaknesses:**

Minor comments/typos:
- Page 2 line 71: "an important one being that the probability mass of any region close to the origin is proportional to its geometric measure" -> "an important one being that the probability mass of any region close to the origin is roughly proportional to its geometric measure".
- Page 5, line 198: Shouldn't the title of the section be "Testing properties of isotropic strongly log-concave distributions"?
- Page 5, line 214: "and runs and in time poly(...)" -> "and runs in time poly(...)"
- Page 5, line 235: There should be a comma between \tau and \delta.
- Page 7, line 307: "Each of the failure events will have probability at least $\delta'$ " -> "Each of the failure events will have probability at most $\delta'$ ".
- Page 8, line 322: "under theempirical distribution" -> "under the empirical distribution".

The following relevant references seem to be missing:
- [1] Sitan Chen, Frederic Koehler, Ankur Moitra, Morris Yau, "Classification Under Misspecification: Halfspaces, Generalized Linear Models, and Connections to Evolvability", NeurIPS 2020.
- [2] Rajai Nasser, Stefan Tiegel, "Optimal SQ Lower Bounds for Learning Halfspaces with Massart Noise", COLT 2022.

**Questions:**

Minor questions:
- Page 5: In Algorithm 1, shouldn't T1 also run with $\delta'$ ?

Optional suggestion:
- Did the authors consider the Tsybakov noise model? It is harder than the Massart noise but easier than the agnostic setting, and hence it might be possible to get an efficient tester-learner that achieves the information-theoretically optimal error $\text{opt}+\epsilon$ for halfspaces with Tsybakov noise (of course, under structured marginal distributions such as Gaussian).

**Limitations:**

No concerns regarding potential societal impact of this work.

---

> ### Author Rebuttal · Authors · 2023-08-09
>
> We thank the anonymous reviewer for their constructive comments and suggestions, as well as for alerting us to a number of typos (including one within Algorithm 1) and relevant references. We also thank the reviewer for pointing to an interesting open direction regarding whether our techniques would also provide information-theoretically optimal results for the Tsybakov noise model. We have not closely considered this model so far, but agree that it would be interesting.

---

> > ### Comment · Reviewer_Vb7c · 2023-08-16
> > **Reply to rebuttal by authors**
> >
> > Thank you very much for your reply. My assessment of the paper remains positive.

---

### Official Review · Reviewer_pNna · 2023-07-06

**Soundness:** 4 excellent
**Presentation:** 4 excellent
**Contribution:** 3 good
**Rating:** 7
**Confidence:** 3

**Summary:**

This papers gives an efficient algorithm for learning halfspaces under the testable learning framework of Rubinfeld and Vasilyan (STOC'23) and facing either Massart or agnostic noise. In this setting, the algorithm is given some reference marginal distribution $D^*$ (which is assumed to be isotropic and strongly log-concave), and it may choose to "reject" (asserting that the actual marginal is different from $D^*$) instead of outputting a hypothesis. Naturally, the learner needs to satisfy the following two conditions:
- Completeness: When the marginal is indeed $D^*$, the learner does not reject w.h.p.
- Soundness: The probability that the learner outputs an insufficiently accurate hypothesis is low. Here, sufficient accuracy means achieving either $\mathsf{opt} + \epsilon$ (under Massart noise) or $\tilde O(\mathsf{opt}) + \epsilon$ error (in the agnostic setting), where $\mathsf{opt}$ is the loss of the optimal halfspace and $\epsilon > 0$ is a parameter.

The solution is built upon the nonconvex optimization approach to learning halfspaces under noise in the literature. The key property for this approach to succeed is that when some appropriate loss function is minimized, all the stationary points are reasonably close to the true parameter. The crux of the current work is then to identify certain testable properties of the marginal under which the above argument goes through, so that we either get a good learning guarantee, or obtain a witness for that the marginal is not $D^*$.

**Strengths:**

The paper studies a fundamental learning theory problem (i.e., learning halfspaces) in the newly introduced testable learning setup. The results are strong and comprehensive, and the solution is nontrivial and requires several novel ideas. The paper is very nicely written and well-strutured, and the main paper contains sufficient details (including the "technical overview" section) for the reader to appreciate the high-level ideas behind the work.

Despite the few weaknesses discussed below, I found the paper a strong submission that should be accepted.

**Weaknesses:**

- The hypothesis class is restricted to homogeneous halfspaces (without a bias).
- In the agnostic case, the error bound can be higher than the optimal error by a constant factor.

**Questions:**

Echoing the weaknesses part, what are the technical hurdles that prevent the current approach from handling non-homogeneous halfspaces and achieving the "$\mathsf{opt} + \epsilon$"-type error guarantee?

**Limitations:**

This is a theory paper and its limitations are in the assumptions made by the problem setting as well as the main results, e.g., the restriction to halfspaces, the noise model, and that a single reference marginal distribution is provided. These are clearly stated in the paper as well as the separate "Limitations and Future Work" section.

---

> ### Author Rebuttal · Authors · 2023-08-09
>
> We wish to thank the anonymous reviewer for their constructive feedback and for appreciating our results!
>
> The problem of designing efficient tester-learners for non-homogeneous halfspaces is an interesting open question. Our approach does not immediately apply to this problem, because we crucially make use of the geometric properties of strongly log-concave distributions around the origin and for that we require the bias to be zero. However, prior work in the distribution specific setting [1], has proposed efficient learning algorithms and it is conceivable that their ideas could be relevant to the testable learning framework.
>
> In the agnostic setting, achieving $\mathrm{opt}+\epsilon$ in fully polynomial time has been shown to be impossible in the statistical query framework (see [2], [3]) or under cryptographic assumptions (see [4], [5]) even in the distribution specific setting (i.e., assuming the marginal to be the standard Gaussian). Therefore, prior work on the distribution specific setting has focused on constant factor approximation algorithms (as well as approximation schemes). The specific technical reason for which our techniques fail to achieve the optimal guarantee is described in lines 166-171 of our submission and amounts to the amplification of the estimation error of the optimal weight vector (e.g., according to Proposition 4.4). In the agnostic setting, the estimation error cannot be chosen to be arbitrarily small, but is instead proportional to $\mathrm{opt}$ (see Lemma 5.2).
>
>
>
> [1] Diakonikolas, I., Kontonis, V., Tzamos, C. &  Zarifis, N.. (2022). Learning General Halfspaces with Adversarial Label Noise via Online Gradient Descent. ICML 2022.
>
> [2] Diakonikolas, I., Kane, D.M., & Zarifis, N. (2020). Near-Optimal SQ Lower Bounds for Agnostically Learning Halfspaces and ReLUs under Gaussian Marginals. NeurIPS 2020.
>
> [3] Goel, S., Gollakota, A., & Klivans, A.R. (2020). Statistical-Query Lower Bounds via Functional Gradients. NeurIPS 2020.
>
> [4] Diakonikolas, I., Kane D.M., & Ren, L, (2023). Near-Optimal Cryptographic Hardness of Agnostically Learning Halfspaces and ReLU Regression under Gaussian Marginals. ICML 2023.
>
> [5] Tiegel, S. (2023). Hardness of Agnostically Learning Halfspaces from Worst-Case Lattice Problems. COLT 2023.

---

> > ### Comment · Reviewer_pNna · 2023-08-14
> > **Thank you for the reply!**
> >
> > I want to thank the authors for the detailed answers. My evaluation of the paper remains positive.

---

### Official Review · Reviewer_oxSi · 2023-07-07

**Soundness:** 3 good
**Presentation:** 3 good
**Contribution:** 2 fair
**Rating:** 3
**Confidence:** 4

**Summary:**

Learning halfspace is a very important problem in machine learning which has been studied extensively. However, generally some distributional assumptions like gaussianity are assumed which in general is difficult to verify. To address this issue, recently Rubinfeld and Vasilyan (STOC 23) have introduced Testable learning framework. Here the primary objective is that if the tester accepts, then the output of the learner is close to OPT + \epsilon (OPT being the optimal error), and when it satisfies the distributional assumptions, the algorithm accepts with high probability. However, when the Gaussian distributional assumption is taken (let's denote this as D^*), it takes $d^{1/\epsilon^2}$ samples, which is also tight. Thus often researchers are interested to design algorithms that have better complexity with respect to $1/\epsilon$, but the error becomes $f(OPT) + \epsilon$ for some function f.

In this work, the authors first design a tester when $D^*$ is isotropic log-concave distribution and the labels are corrupted according to Massart noise (the labels are changed by an adversary with a small probability $\eta$).  Their algorithm runs in polynomial time and has error $OPT + \epsilon$ (Theorem 4.1). Later they design testers for adversarial noise with respect to Gaussian distribution with error $O(OPT) + \epsilon$ (Theorem 1.2).

In Section 4, the authors study the case with Massart noise. The primary idea here is to minimize a non-convex smooth surrogate loss (4.1) such that its stationary points correspond to halfspaces with small error. The authors first run PSGD on this surrogate loss function to get a set of vectors L such that one vector in L is close to the optimal weight vector. Then they apply localization ideas based upon the region that is an axis-aligned rectangle T, and check if the low degree moments of input distribution D conditioned on T match with D^* conditioned on T. This will ensure that the angular distance of a stationary point w is close to the optimal w^* (Lemma 4.3). To convert closeness in angular distance to closeness in 0-1 loss, they use the fact that the distribution is isotropic strongly log-concave (Proposition 4.4).

Later in Section 5, the authors study the agnostic setting where they will call the algorithm from Section 4 several times, each time with different parameters. The idea is that in the agnostic setting, running the algorithm for only once, the algorithm might only consider points that lie within a region with small probability. This finally gives a tester with error $O(OPT) + \epsilon$  when $D^*$ is isotropic log-concave distribution as well as Gaussian (Theorem 5.1 and Theorem 5.3).

**Strengths:**

The paper gives the first algorithm for testable learning of halfspaces that runs in poly(d, epsilon). The algorithm is very nice. With the complexity pulled down drastically, a proper implementation and experimental results for this algorithm would be possible and it would be nice to see the relevance of the concept of testable learning in various applications.

Also the algorithm can handle both adversarial and massard noise.

**Weaknesses:**

The usefulness of the testable learning model in real life applications is yet to be understood.

**Questions:**

Can this approach be used to design tester-learners for function classes other than halfspaces.



**Limitations:**

It is a pure theoretical work in the paradigm of testable learning - a relatively new concept whose importance is not yet fully confirmed.

---

> ### Author Rebuttal · Authors · 2023-08-09
>
> We thank the anonymous reviewer for their feedback and comments. While our results are indeed of theoretical nature, we view the testable learning framework as an important step towards bridging theory with practice, since it removes a significant part of the modeling assumptions typically required to achieve provable guarantees for agnostic learning algorithms (i.e., we obtain provable guarantees that do not require any assumptions about the marginal distribution!). Further, the framework provides a new and theoretically well-founded way for a learning algorithm to say “I do / do not know,” which is a a problem of great relevance for reliable machine learning in general. We view the specific problem considered in this paper, namely attaining $O(\mathrm{opt})$ tester-learners for halfspaces, as one important step towards building the foundations of this framework.
>
> Designing tester-learners for other function classes (e.g., neurons with different activations) is an interesting open problem. In particular, for the problems of ReLU and of sigmoid regression, there are efficient algorithms that work in the distribution specific setting, but whether they can be extended to the testable learning framework remains open.

---

> > ### Comment · Reviewer_oxSi · 2023-08-16
> >
> > I have read the reply. Thank you for the detailed response.

---

### Official Review · Reviewer_GfRi · 2023-07-12

**Soundness:** 4 excellent
**Presentation:** 4 excellent
**Contribution:** 3 good
**Rating:** 7
**Confidence:** 4

**Summary:**

This work provides an efficient algorithm for testably learning halfspaces, extending the frontier of the recently introduced testable learning which does not assume anything about the given data distribution. Specificially, the setting is as follows: the target distribution is standard Gaussian (or any fixed strongly log-concave distribution) and the label noise is Massart or adversarial (agnostic).

The main result is two-fold.
1. For Massart noise, if the target distribution is strongly log-concave, the paper proves an algorithmic guarantee that testably learns halfspaces up to $opt + \epsilon$ error and runs in $poly(d,1/\epsilon,1/(1-2\eta), \log(1/\delta)$ time.
2. For agnostic learning,  if the target distribution is strongly log-concave, the guarantee is that the algorithm testably learns halfspaces up to $O(k^{1/2} opt^{k/(k+1)} + \epsilon$ error and runs in "roughly" $poly(d^k,1/\epsilon^k, \log^k(1/\delta)$ time (ignoring some logarithmic factors). One can strengthen this result if the target distribution is standard Gaussian. Then the algorithm testably learns halfspaces up to $O(opt) + \epsilon$ error and runs in $poly(d,1/\epsilon, \log(1/\delta)$ time, a result that matches previous non-testable learning results for halfspaces.

The methodology borrows two algorithmic ideas and strengthens them. One is the algorithmic idea that runs (nonconvex) SGD on a convex surrogate (ramp function) for the 0-1 loss. Originally, given some distributional assumption, this approach would yield a hypothesis found from an approximate stationary point. In this paper, the authors check if such property is satisfied for the unknown distribution of the testable learning setting, leading to develop a three-stage testing procedure for strongly log-concave distributions. Here, the second algorithmic idea of moment matching kicks in. For the band $T$ s.t. $|\langle w,x \rangle| \le \sigma$, tests are ran on the empirical distribution conditioned on $T$ to check moments matching those of the target distribution on $T$.

**Strengths:**

- This work extends upon the recently introduced testable learning in one of the fundamental problems of learning, i.e., learning halfspaces. I believe the topic is of good importance as testable learning yields more practicality to learning algorithms. In that sense, the work studies and provides strong algorithmic result to an important problem.
- The techniques used in the paper utilize two previous algorithmic ideas (convex surrogate SGD + moment-matching to fool) and neatly tie these two ideas together to a testable algorithm. The algorithmic techniques are also practical.
- The tests are ran on the distribution conditioned on the band $T$. With this "trick", the paper manages to change weak additive guarantees to strong multiplicative ones.
- The results are technically strong and presentation is clear.
- The agnostic learning algorithm only modifies the Massrt one slightly, which is neat, though this may be more of contribution of previous work than that of this work.

**Weaknesses:**

No notable weaknesses, but refer to Questions for a potential undecided one.

**Questions:**

- For agnostic testable learning for strongly log-concave distributions, the guarantees are weaker. Is there any inherent difficulty (e.g., lower bounds or conjectured hardnesses) that may explain the weaker guarantees? What is the intuitive reason why Massart and agnostic differ in results?
- (Typo) L201 "stated here as Proposition A.3": Should this be 3.1 instead of A.3?

**Limitations:**

No limtation addressed.

---

> ### Author Rebuttal · Authors · 2023-08-09
>
> We wish to thank the anonymous reviewer for their feedback and for appreciating our work!
>
> Regarding the reviewer’s question, prior work (e.g., [1], [2], [3], [4]) has provided evidence (in terms of statistical-query or cryptographic lower bounds) that achieving $\mathrm{opt}+\epsilon$ for the problem of agnostically learning halfspaces, even assuming Gaussian marginals, is hard (i.e., requires exponential dependence on $1/\epsilon$). As a result, recent prior work on the distribution specific setting has focused on providing efficient constant factor approximation algorithms (and polynomial time approximation schemes). From the perspective of the techniques we use, the reason that we get qualitatively different results for the agnostic case than in the Massart noise case is as follows (see also lines 166-171):
> 1. By finding a stationary point of the surrogate loss, we estimate the optimal weight vector up to some error.
> 2. The error of our weight vector estimate is amplified (according to Proposition 4.4, where $\theta$ is the error of our weight vector estimate).
> 3. In the Massart noise case, we can make $\theta$ arbitrarily small by using a polynomial number of resources (see Lemma 4.3), while in the agnostic noise case, $\theta$ is proportional to $\mathrm{opt}$ (see Lemma 5.2).
> 4. Therefore, in the final error, the amplified estimation error can be made arbitrarily small in the Massart noise case, but contains a function of $\mathrm{opt}$ in the agnostic noise case.
>
> We also note that, in the agnostic case, we obtain $O(\mathrm{opt})$ only for Gaussian target marginals for technical reasons and we leave the extension of such a result to broader families of target marginals for future work (see also lines 88-93).
>
>
> [1] Diakonikolas, I., Kane, D.M., & Zarifis, N. (2020). Near-Optimal SQ Lower Bounds for Agnostically Learning Halfspaces and ReLUs under Gaussian Marginals. NeurIPS 2020.
>
> [2] Goel, S., Gollakota, A., & Klivans, A.R. (2020). Statistical-Query Lower Bounds via Functional Gradients. NeurIPS 2020.
>
> [3] Diakonikolas, I., Kane D.M., & Ren, L. (2023). Near-Optimal Cryptographic Hardness of Agnostically Learning Halfspaces and ReLU Regression under Gaussian Marginals. ICML 2023.
>
> [4] Tiegel, S. (2023). Hardness of Agnostically Learning Halfspaces from Worst-Case Lattice Problems. COLT 2023.

---

> > ### Comment · Reviewer_GfRi · 2023-08-14
> > **Response to Authors**
> >
> > I have read the reply. Thank you for the detailed response.

---

### Decision · Program_Chairs · 2023-09-21

**Decision:**

Reject

**Comment:**

As was acknowledged in the introduction of submission 14165, the results of this paper were dominated by the results of that submission.  The program committee decided that a paper should not be published in the same conference in the same year as a paper that dominates its results.